# Coupling of autism genes to tissue-wide expression and dysfunction of synapse, calcium signalling and transcriptional regulation

Jamie Reilly[1]*, Louise Gallagher[2,3], Geraldine Leader[4], Sanbing Shen[1,5]*

**1** Regenerative Medicine Institute, School of Medicine, Biomedical Science Building, National University of Ireland (NUI) Galway, Galway, Ireland, **2** Discipline of Psychiatry, School of Medicine, Trinity College Dublin, Dublin, Ireland, **3** Trinity Translational Medicine Institute, Trinity Centre for Health Sciences—Trinity College Dublin, St. James's Hospital, Dublin, Ireland, **4** Irish Centre for Autism and Neurodevelopmental Research (ICAN), Department of Psychology, National University of Ireland (NUI) Galway, Galway, Ireland, **5** FutureNeuro Research Centre, Royal College of Surgeons in Ireland (RCSI), Dublin, Ireland

* J.Reilly12@nuigalway.ie (JR); Sanbing.shen@nuigalway.ie (SS)

**Data Availability Statement:** All data used in this paper are publicly available. The file for median GTEx expression data can be found here (https://gtexportal.org/home/datasets). The data from HPA

## Abstract

Autism Spectrum Disorder (ASD) is a heterogeneous disorder that is often accompanied with many co-morbidities. Recent genetic studies have identified various pathways from hundreds of candidate risk genes with varying levels of association to ASD. However, it is unknown which pathways are specific to the core symptoms or which are shared by the co-morbidities. We hypothesised that critical ASD candidates should appear widely across different scoring systems, and that comorbidity pathways should be constituted by genes expressed in the relevant tissues. We analysed the Simons Foundation for Autism Research Initiative (SFARI) database and four independently published scoring systems and identified 292 overlapping genes. We examined their mRNA expression using the Genotype-Tissue Expression (GTEx) database and validated protein expression levels using the human protein atlas (HPA) dataset. This led to clustering of the overlapping ASD genes into 2 groups; one with 91 genes primarily expressed in the central nervous system (CNS geneset) and another with 201 genes expressed in both CNS and peripheral tissues (CNS+PT geneset). Bioinformatic analyses showed a high enrichment of CNS development and synaptic transmission in the CNS geneset, and an enrichment of synapse, chromatin remodelling, gene regulation and endocrine signalling in the CNS+PT geneset. Calcium signalling and the glutamatergic synapse were found to be highly interconnected among pathways in the combined geneset. Our analyses demonstrate that 2/3 of ASD genes are expressed beyond the brain, which may impact peripheral function and involve in ASD co-morbidities, and relevant pathways may be explored for the treatment of ASD co-morbidities.

can be accessed via Bioconductor in R, using the code provided in the supplementary information. The networks from Huri can be accessed from NDEX (http://www.ndexbio.org/#/user/69e7b21d-8981-11ea-aaef-0ac135e8bacf) Information from the scoring systems used can be found in the supplementary information sections of their respective publications. Differentially Expressed genes can be found in supporting information from respective publications and in Geo2r (GSE42133 for Pramparo, GSE28521 for Voineagu). Code is provided for data access to HPA and GeneOverlap and generating figures. Genelists were obtained from pysgenet(http://www.psygenet.org/web/PsyGeNET/menu), Harmonizome(https://maayanlab.cloud/Harmonizome/dataset/GWASdb+SNP-Disease+Associations) and the supplementary information from respective publications.

**Funding:** Funding was provided by Science Foundation Ireland (Grant 13/IA/1787 to S.S. and L.G., 16/RC/3948 to FutureNeuro), and National University of Ireland Galway (Grant RSU002 to S. S.) for funding the research.

**Competing interests:** The authors have declared that no competing interests exist.

**Abbreviations:** A1C, Primary auditory cortex; AMY, Amygdaloid complex; Ary, Arrythmia; ASD, Autism Spectrum Disorder; BP, Bipolar Disorder; CBC, Cerebellar cortex; CNS, Central Nervous System; DAMAGES, Disease-Associated Mutation Analysis Using Gene Expression Signatures; DFC, Dorsolateral prefrontal cortex; DNA, Deoxyribonucleic Acid; EXAC, Exome Aggregation Consortium; GABA, Gamma-Aminobutyric Acid; GTEx, Genotype-Tissue Expression; HIP, Hippocampus; HPA, Human Protein Atlas; Huri, Human Reference Interactome; IBD, Inflammatory bowel disease; IPC, Posterior inferior parietal cortex; iPSC, Induced Pluripotent Stem Cell; ITC, Inferolateral temporal cortex; M1C, Primary motor cortex; MAPK, Mitogen Activated Protein Kinase; MD, Mediodorsal nucleus of thalamus; MDD, Major Depressive Disorder; MFC, Medial prefrontal cortex; OFC, Orbital frontal cortex; ORA, Overrepresentation Analysis; PT, Peripheral Tissue; S1C, Primary somatosensory cortex; SFARI, Simons Foundation for Autism Research Initiative; STC, Posterior(caudal) superior temporal cortex; STR, Striatum; T1D, Type 1 diabetes; T2D, Type 2 diabetes. SCZ–Schizophrenia; V1C, Primary visual cortex; VFC, Ventrolateral prefrontal cortex.

## Introduction

Autism Spectrum Disorder (ASD) is a heterogeneous and complex neurodevelopmental disorder [1], with core features including stereotypical behaviours and impaired social and communication skills, and with various comorbidity. The CNS comorbidity of ASD includes epilepsy, sleeping disorders [2], intellectual disabilities, language delay, anxiety and hyperactivity [3, 4], and the peripheral comorbidity includes gastrointestinal, metabolic disorders, auto-immune disorders, tuberous sclerosis, attention-deficit hyperactivity disorder, and sensory problems associated motor problems [2, 5–8]. It appears that genetic heterogeneity and environmental factors impact not only the severity of ASD, but also the presence and severity of comorbid disorders [9]. However, it is unknown why different individuals display overlapping core symptoms and/or different comorbidities.

With the advent and increasing availability of DNA sequencing over the past decade, much has been uncovered about the genetics of ASD [10], which include *de novo* events [11–17], mosaic mutations [18] and gene dosage changes resulted from copy number variations [19–25], as well as epigenetic/transcriptome changes with no apparent genetic alterations [26]. As a result, hundreds to thousands of ASD risk factors have been identified by different studies, suggesting that ASD is a multi-genetic disorder and each has small effects in terms of ASD population. This presents a huge challenge to develop ASD diagnosis or treatment. Meanwhile, little is known about which set of genetic factors links to peripheral comorbidity, and it is therefore crucial to decipher factors/pathways which are associated with the comorbidities.

The genetic studies have allowed formation of ASD databases for investigations [27, 28]. The early network analyses using the SFARI database have identified ASD pathways of abnormal synaptic function, chromatin remodelling and ion channel activity [29] which are highly connected by MAPK signalling and calcium channels, with some genes associated with cardiac and neurodegenerative disorders [30]. This was carried out before the scoring system from SFARI became available. In addition, the SFARI list of ASD genes has risen from 680 in 2016 to 1053 in January 2019. Furthermore, independent scoring systems have become available and suggested additional genes with significance to ASD, from either sequencing thousands of individuals across the globe [31], or using existing interaction databases in conjunction with SFARI database [32, 33], or employing machine learning on datasets [32, 34]. This suggests that another round of ASD pathway analysis is due.

Our hypotheses were that the high ASD candidates would recur in different scoring systems, and that comorbidities in ASD would involve expression of risk genes in relevant tissues/organs. Therefore, in this study, we focused on the identification of the overlapping genes in the updated SFARI database and autistic genes shortlisted by the majority of the third-party scoring systems as summarised in Table 1. To explore the biological context of these genes, we first examined their expression using the GTEx database to find out if they were transcribed not only in the brain but also in other peripheral tissues, and then used the human protein atlas (HPA) dataset to verify protein expression levels. We also explored tissue specific networks from human reference interactome (Huri) [35] to see if any ASD candidates interacted with genes in these networks.

Our analyses suggest that a third of ASD risk genes (CNS geneset) is specifically expressed in the CNS, which are involved in brain development, synaptic function and ion transport, whereas the majority of ASD factors are highly expressed in both CNS and peripheral tissues (CNS+PT geneset), with pathways of brain development, chromatin organisation and gene regulation, which may account for ASD peripheral comorbidity.

**Table 1. Overview of five datasets and the independent scoring systems used to shortlist ASD genes for overlapping analysis.**

| Score name | Data Source | Starting genes | Score Threshold | Shortlisted genes | References |
|---|---|---|---|---|---|
| EXAC | Exome sequences from 60,706 individuals | 15735 | pLi >0.9 with >90% negative effect upon mutation | 3126 | Lek et al. [31] |
| SFARI | SFARI GENE (Jan 2019) | 1053 | All genes recorded in Jan 2019 | 1053 | Gene.SFARI.org (Jan 2019) |
| Krishnan | SFARI, OMIM, GAD, HUGE (up to 2013) | 25825 | q value < 0.05 | 3225 | Krishnan et al. [32] |
| Duda | Microarray data from human, mouse and rat. Protein interaction databases (MIPS, BIOGRID, MINT, IntAct) | 21115 | Top 10th percentile | 2111 | Duda et al. [34] |
| Zhang | Mouse CNS Microarray Data, Genes homologous to human | 15950 | Positive DAMAGE score (D>0) | 7189 | Zhang et al. 2017 |

## Methods

### Datasets and shortlisting of the ASD risk factors

Five datasets were chosen as the starting point of this study [28, 31–34], and high-ranking genes were shortlisted from each dataset with defined criteria (Table 1, Fig 1). For the Exome Aggregation Consortium (EXAC) with exome sequencing data from ~60,000 cases (S1 Table), a high intolerance to mutation of pLi $\geq$ 0.9 was applied, where pLi indicated the level of intolerance to mutations in a given gene, with many containing loss of function variants. In the Krishnan's geneset (S2 Table) created using human disease databases of GAD, OMIN, HUGE and SFARI in December 2013, along with a brain-specific functional interaction network, a q-value of $\leq$0.05 was used, where q-value was the probability of the gene being an ASD risk

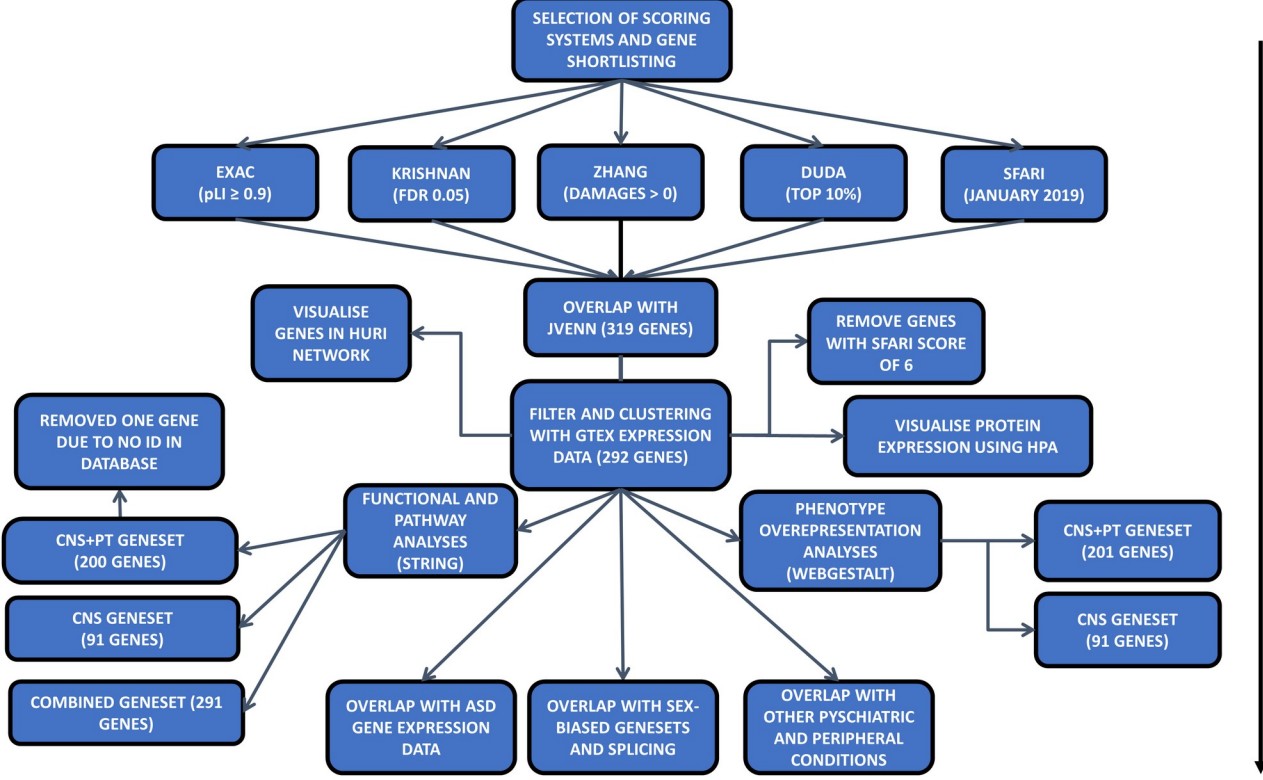

**Fig 1. Flowchart of the analysis for the paper: Arrow indicates the steps of each analysis.**

candidate after multiple testing for false positivity. For the Zhang's geneset (S3 Table) made using CNS microarray expression data from six brain regions derived from mice and validated using data from exome sequencing studies [11, 14, 17, 29, 36, 37], mutations of the human homologues with a (DAMAGE) score of D≥0 were shortlisted, where positive D-score was a measure of a mutation's likeliness to be associated with ASD. For the Duda's list (S4 Table) which was created from *de novo* mutation analysis, protein-protein interaction and phenotype information, the top 10% of genes in the list were selected, as this was used as a cut off in the original publication. Finally, for the SFARI collection, all genes up to January 2019 were included (S5 Table).

## Overlap of shortlisted genes with SFARI

After shortlisting of high-ranking genes from each dataset, the Jvenn program was applied to identify the common ASD risk genes among the 5 sources [38]. A list of 519 genes, which were overlapped among 4 of 5 sources, was extracted (Table 1, Fig 1). Among them, 319 genes appeared in the SFARI database were taken forward for expression and enrichment analyses.

## Filtering using gene expression analysis data

Since ASD has a wide array of peripheral co-morbidities, we believe that some ASD genes are expressed in peripheral tissues. To explore this possibility, mRNA expression data were downloaded from the GTEx consortium (v7) in the form of median TPM (Transcripts Per Million) values from all tissues [39]. We excluded low expression genes based on the average of the median (TPM<3) in both CNS and PT groups. Additionally, we removed genes with a SFARI score of 6, which were not likely to be associated with ASD. The expression data was applied to the 319 selected genes and uploaded to Morpheus (https://software.broadinstitute.org/morpheus) to generate heatmap (with settings for clustering: hierarchical, euclidean distance, linkage method complete, clustering based on columns). Genes with extremely low levels of mRNA expression, at an average TPM≤3 in both the brain and peripheral tissues, were excluded from the subsequent analyses (S8 Table).

## Overview of protein expression levels

As an additional level of verification, we used HPA data (v19.3) obtained via HPA analyze [40], a Bioconductor program that runs in R, to assess protein expression levels across multiple tissues among the two genesets, and used ggplot2 [41] to visualise the data.

## Tissue specific interaction networks

The expression of ASD genes in other tissues indicates that they may interact with other factors in tissue-specific networks. To explore this, we used tissue-specific networks generated from Huri to see if ASD candidate genes were present in other networks, and if they had any interacting partners within these networks.

## Functional enrichment analysis of the final geneset

The final list of ASD common risk factors was analyzed through STRING program for pathway analyses, except for SHANK3 that was misidentified as HOMER2 in the current human database of STRING v11. The resulting GO Terms (Biological Processes, Cellular Components, and Molecular Function) and KEGG pathways were downloaded. The same list was also loaded into Cytoscape [42] to identify sub-clusters of genes in interaction network.

### Analysis of genesets for co-morbid phenotypes

To examine potential association of the ASD genesets with occurring co-morbidities, an over-representation analysis (ORA) was carried out using the tool WebGestalt [43] to assess other co-morbid conditions linking to the ASD genesets. The Human Phenotype Ontology database was used for the analysis [44]. The top 50 terms were used as a cut off to balance between the co-morbidities reported in ASD and to ensure that the final lists are not too broad and overly diluted.

### Comparison of shortlisted ASD geneset with ASD expression studies

To examine the utility of the ASD geneset in the literature of ASD gene expression studies, we compared the ASD geneset with DEGs reported from post-mortem brain [45], blood [46–48] and GI tissue [46, 49], as well as iPSC-derived cell models [50–55], to see if any of the ASD genes were significantly up or downregulated. The DEGs were obtained using autism versus control group with FDR (adj p-value) at 0.05, except for Voineagu [56] and Pramparo [47], which we used Geo2R [57] using autism versus controls as groups to obtain DEGS at 0.05 FDR (adj p-value).

### Expression of genes across brain development and sex-bias

We used CSEA [58] tool to check for enrichment of our genes for human gene expression across developmental periods and brain tissues in the Brainspan dataset. We also used the genesets from the publications [59, 60] to see if any of our shortlisted genes had sex-bias expression in prenatal stages [59], or if there was sex-specific splicing in our geneset [60] using the bioconductor package GeneOverlap [61].

### Comparison with psychiatric and peripheral diseases

Our functional analyses of the ASD genesets showed enrichment for processes in areas relating to cardiac function and insulin. In addition, it is known that many ASD risk genetic factors share molecular pathways with other psychiatric conditions. To future explore this, we downloaded genesets for schizophrenia (SCZ), bipolar disorder (BP), and major depressive disorder (MDD) from Psygenet [62]. We also downloaded genelists from Harmonizome database [63, 64] for 4 peripheral conditions of arrythmia (ARY), type 1 (T1D) and type 2 (T2D) diabetes based on our results, and inflammatory bowel disease (IBD) based on co-morbidity of gastrointestinal issues with ASD for comparison of ASD risk factors in the current study.

## Results

To explore the hypothesis that ASD candidate genes were recurrent across different scoring systems, we selected the SFARI dataset and four other published scoring systems for the current study (Table 1, Fig 1). Consequently, we selected 3126 genes out of 15735 from the EXAC dataset with pLi ≥ 90% chance of intolerance to loss of function [31], 3225 genes from Krishnan's data [32] based on q<0.05, 7189 genes out of 15950 from Zhang's data [33] with a positive DAMAGE score (D>0), and 2111 genes from the top 10% of the 21115 genes in the Duda's data [34], and all 1053 SFARI genes as of Jan 2019 [28].

These shortlists were subsequently analysed by Jvenn web tool for overlapping ASD genes [38], and 114 genes were found to be shared by all five shortlists (Fig 2, S6 Table). Considering that some well-known ASD genes such as *SHANK3* and *CHD8* were excluded from the 114 geneset, we adjusted to include the genes that were overlapped in 4 of the 5 scoring systems, which resulted in a total to 519 ASD risk genes. Further examination of the 519 genes for the

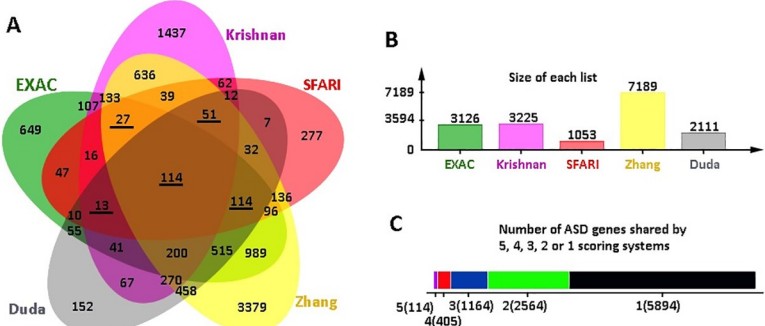

**Fig 2. Shortlisting of common ASD factors.** (A) JVenn Diagram of all overlapping genes from the scoring systems used. Underlined numbers indicate sets overlapped with SFARI. (B) Size of each set of shortlisted genes. (C) The number of genes that are shared by or specific to the lists.

presence in the SFARI dataset has narrowed down the list to 319 SFARI genes, which were highly ranked in >3 of 4 other independent studies besides the SFARI dataset (Table 1, Fig 1).

## Expression pattern of ASD genes

To assess the gene expression across different tissues, we next examined body-wide expression of the 319 genes using the GTEx dataset containing standardized mRNA expression in units of TPM. This further reduced the ASD genes to 292 genes to filter out low abundance genes with TMP≥3 in both CNS and PT (S7 Table).

The expression levels of the 292 genes were presented in the Heatmap, with high expression coloured in red, low expression in green and no expression in grey. Analysis of the 292 genes with GTEx (S8 Table) showed that the ASD genes were clustered into 2 groups (Fig 3); 91 genes were mainly expressed in the CNS with TMP<3 in PT, whereas the remaining 201 genes were ubiquitously expressed in both the CNS and PT.

A similar expression profile for proteins was observed in HPA data (Figs 4 and 5), whereas CNS geneset showed protein expression mostly in the CNS (Fig 4), the CNS+PT geneset showed protein expression in both the CNS and other tissue types (Fig 5). We also identified that some of the ASD factors in the CNS+PT geneset (S1–S19 Figs) were indeed interacting with other proteins in tissue-specific networks (Table 2). Interestingly HDAC4 (S6) in the colon and STX1A (S15) in the pituitary gland were found to be tissue-specific genes in these organs.

## Enriched neurodevelopment, synaptic function, and ion transport in the CNS geneset

To assess the biological context of ASD factor, the STRING (v 11) program was used to analyse two groups of the ASD factors, respectively [65]. The 91 CNS-specific geneset gave rise to 305 "*Biological Processes*" (S10 Table), 73 "*Cellular Components*" (S11 Table) and 62 "*Molecular Function*" (S12 Table). Go term results revealed an enrichment for neuronal development and synaptic function in the CNS geneset (Fig 6A).

We also visualized the interaction network using Cytoscape (V3.7). Among the 91 CNS genes (Table 3), 47 (colored in blue) were involved in "Nervous system development" (FDR = 1.39E-18), 31 (colored in red) were linked to "Trans-synaptic signalling" (FDR = 2.41E-25), and 30 were associated with "Ion transmembrane transport" (colored in yellow, FDR = 1.55E-14, Fig 6A), which included 6 calcium channels (*ATP2B2*, *CACNA1A*,

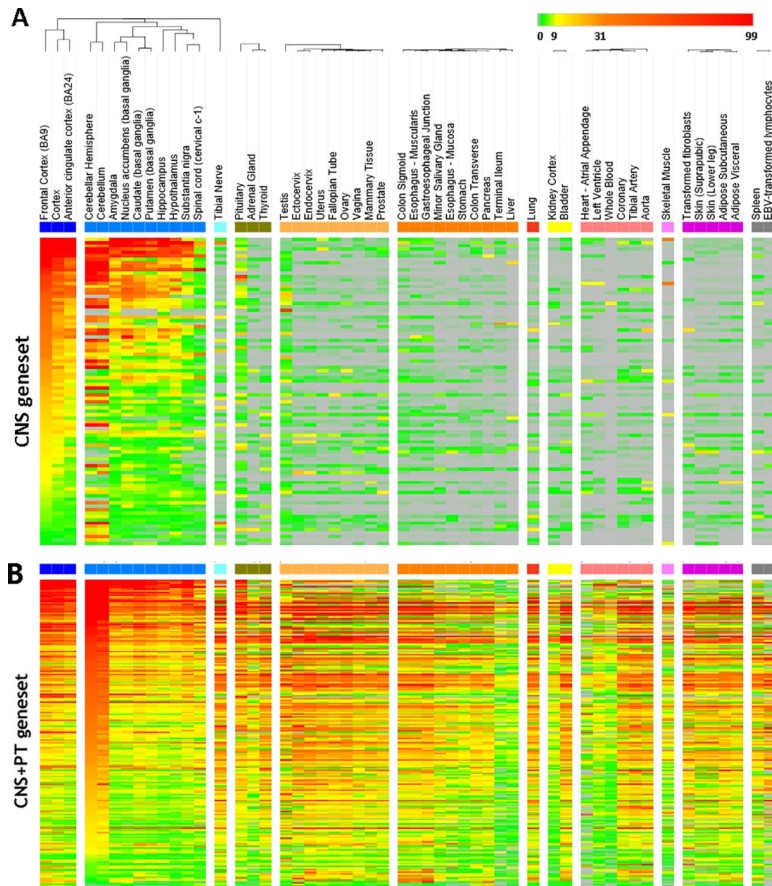

**Fig 3. Heatmap of 292 genes after filtering for low expressed genes (TPM<3).** The tissues were indicated on the top. Scale bar on top right corner showed the scale of mRNA expression for each gene: grey is TPM = 0, green is 0–3, yellow is 9–31, orange is 31–99, red is > 99. (A) The 91 genes in the CNS geneset have an average median of TMP<3 in the peripheral tissues. (B) The 201 CNS+PT genes were expressed both CNS and peripheral tissues with TMP>3.

*CACNA1B*, *CACNA1D*, *CACNA1G*, *CACNA2D3*), 3 sodium channels (*SCN1A*, *SCN2A*, *SCN8A*), 4 potassium (*HCN1*, *KCND2*, *KCNQ2*, *KCNQ3*) channels, 6 glutamatergic receptors (*GRIA1*, *GRID1*, *GRIK2*, *GRIN1*, *GRIN2A*, *GRIN2B*), 5 GABAergic receptors (*GABRA1*, *GABRA3*, *GABRA4*, *GABRA5*, *GABRB3*) and 5 transporters (*SLC12A5*, *SLC1A2*, *SLC24A2*, *SLC30A3*, *SLC4A10*).

Consistently, the synapses (42/91 genes, FDR = 4.06E-30), neuronal projection (44/91, FDR = 9.54E-28) and ion channel complex (24/91, FDR = 7.59E-22, Table 3) were enriched in "*Cell Components*". The ion-gated channel activity (24/91, FDR = 5.78E-19) and neurotransmitter receptor activity (13/91, FDR = 3.60E-13) including glutamate (7/91, 2.95E-09, *GRIA1*, *GRID1*, *GRIK2*, *GRIN1*, *GRIN2A*, *GRIN2B*, *GRM1*) and GABA receptor activity (6/91, 4.12E-08, *GABBR2*, *GABRA1*, *GABRA3*, *GABRA4*, *GABRA5*, *GABRB3*), were the most significant "*Molecular Functions*". The "*KEGG pathway*" analyses (S13 Table) demonstrated that Glutamatergic synapse (12/91, 2.46E-11, *CACNA1A*, *CACNA1D*, *DLGAP1*, *GRIA1*, *GRIK2*, *GRIN1*, *GRIN2A*, *GRIN2B*, *GRM1*, *HOMER1*, *SHANK2*, *SLC1A2*), GABAergic synapse (11/91, 3.44E-11, *CACNA1A*, *CACNA1B*, *CACNA1D*, *GABBR2*, *GABRA1*, *GABRA3*, *GABRA4*, *GABRA5*, *GABRB3*, *GAD1*, *SLC12A5*), Calcium signalling pathway (11/91, 1.80E-08, *ATP2B2*, *CACNA1A*, *CACNA1B*, *CACNA1D*, *CACNA1G*, *CAMK2A*, *ERBB4*, *GRIN1*, *GRIN2A*, *GRM1*, *NOS1*) and Circadian entrainment (9/91, 1.74E-08, *CACNA1D*, *CACNA1G*, *CAMK2A*,

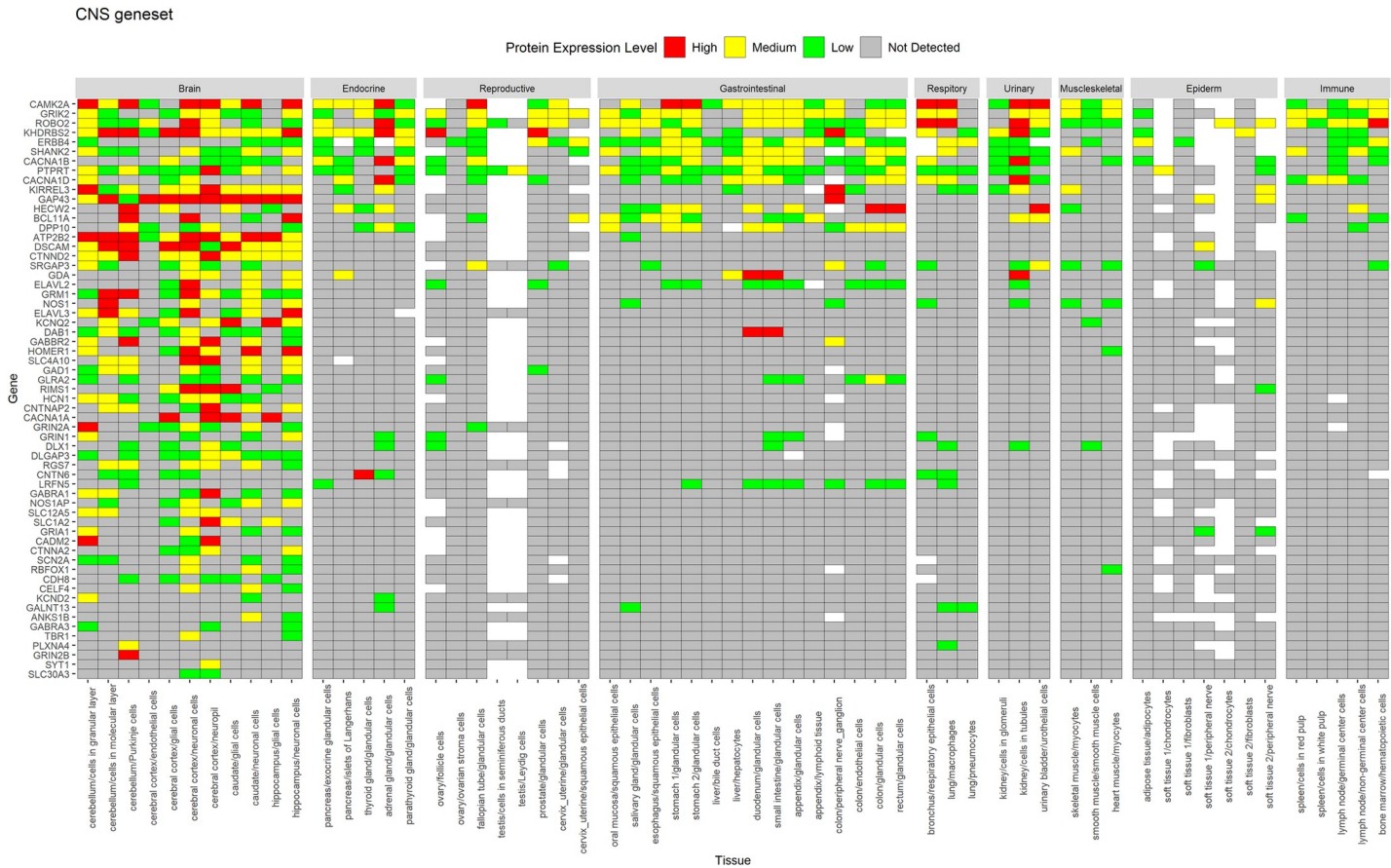

**Fig 4. Protein expression of CNS geneset from HPA across tissues.** Genes are displayed in order of decreasing protein expression.

*GRIA1*, *GRIN1*, *GRIN2A*, *GRIN2B*, *NOS1*, *NOS1AP*) were the top pathways in the CNS geneset (Table 3). Together, these data suggest that the 91 CNS-specific ASD risk factors are involved in regulation of brain development, E/I balance and calcium signalling, which are closely related to the ASD core features, and to the CNS comorbidity such as epilepsy, intellectual disability and sleeping disorders.

## Enriched chromatin organisation and gene regulation in CNS+PT geneset

Analyses of the 200 CNS+PT genes resulted in 546 "*Biological Processes*" (S14 Table), 127 "*Cellular Components*" (S15 Table), and 123 "*Molecular Function*" (S16 Table). Like the CNS geneset, CNS development and synapse are the top pathways in the CNS+PT ASD geneset. This included Nervous system development (72/200 genes in blue, FDR = 7.50E-17, Fig 6B), Modulation of chemical synaptic transmission (20/200, FDR = 1.74E-08) and Trans-synaptic signalling (22/200, FDR = 3.16E-08) as top "*Biological Processes*", Synapse (46/200 in red, FDR = 6.44E-18, Fig 6B) as the most significant "*Cellular Components*", Ion binding (106/200, FDR = 2.09E-08) in the "*Molecular Function*" (Table 3), and Long-term potentiation (6/200, FDR = 0.003), Glutamatergic (7/200, FDR = 0.006), and Dopaminergic synapse (7/200, FDR = 0.011) identified in the "*KEGG*" pathways (Table 3, S17 Table).

However, the most prominent feature of the CNS+PT ASD geneset was transcription regulation, and this included Nucleus (128/200, FDR = 1.73E-14) in the "*Cellular Components*",

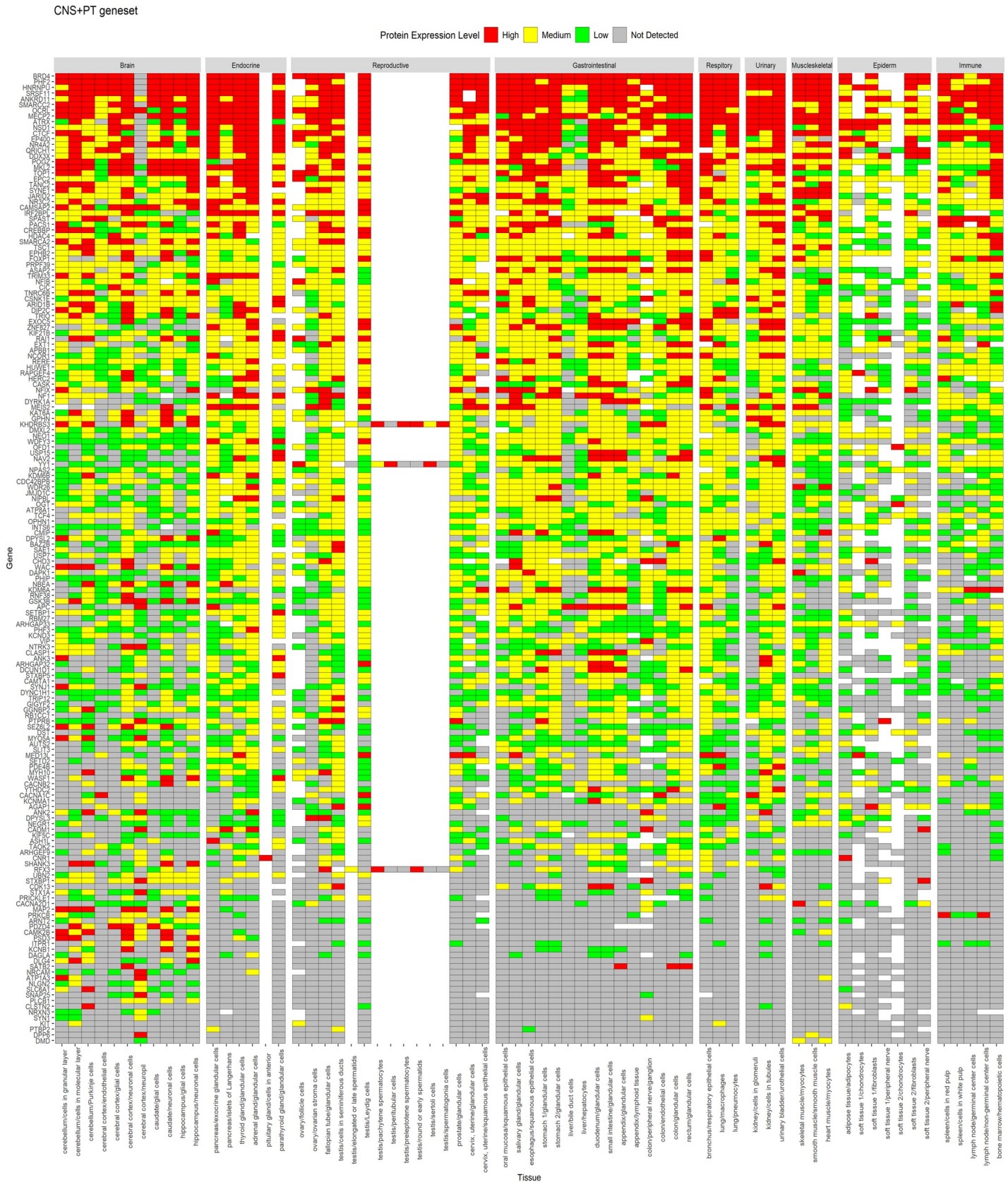

**Fig 5. Protein expression of CNS+PT geneset from HPA across tissues.** Genes are displayed in order of decreasing expression.

**Table 2. Interacting ASD genes in tissue-specific networks.** Genes in bold denote ASD genes (*POGZ* and *ANKRD11*) present in nearly all networks.

| Tissue | ASD genes | Interacting partners |
|---|---|---|
| Heart left ventricle | *TCF4, RFX3, HDAC4,* **POGZ, ANKRD11**, *RERE, YY1, CMIP, QRICH1* | *MYH7B, TCF24, ASB15, SNRPC, TRIM54, FSD2, FHL2* |
| Heart-left atrial appendage | *HDAC4,* **ANKRD11**, *TCF4, CMIP, QRICH1* | *TRIM54, ASB15, MYH7B, TNNI1, TCF24, FSD2* |
| Artery-tibial | **ANKRD11** | *NOV* |
| Small Intestine-terminal ileum | *TCF4,* **POGZ**, *CLSTN3, NSD1* | *A1CF, AOC1, BCL2L15, AGR2, OLFM4* |
| Pancreas | *MEIS2, DAGLA, STX1A, TSC1, ARNT2, TCF4,* **POGZ**, *CLSTN3,* **ANKRD11** | *FAM136A, LHFPL5, PNLIPRP1, SERP1, ENKD1, SIM1, NEUROD, BCL2L15, BANF2, A1CF, OLFM4, TMEM97* |
| Stomach | *PRICKLE1,* **ANKRD11**, *TCF4,* **POGZ**, *NSD1* | *BCL2L15, ODAM, AGR2, JRK* |
| Esophagus-mucosa | *DYRK1A, ARNT2, WAC, KATNAL1, TSC1, RNF38, EXOC5, YY1, HDAC4, MEIS2,* **POGZ, ANKRD11** | *DTX2, BICD2, USH1G, TXN, CYSRT1, LGALS7B, LGALS7* |
| Esophagus-muscularis | *USP7* | *ANKS1A* |
| Muscle-skeletal | *NPAS2, TSC1, MEIS2, TCF4, CMIP,* **POGZ, ANKRD11**, *SRSF11, HDAC4, STX1A, QRICH1, CACNB2* | *KRT31, ATG9A, CDR2L, TRIB3, MYF5, FAM222B, FAM166B, BICD2, MEF2A, MEF2C, GOLGA2, SSX2IP, AES, NGLY1, YWHAE, DUPD1, FHL3, TADA2B, TRIM27, CALCOCO2, INCA1, OIP5, LBX1, NFKBID, ASB15, CEP70, USP6, ZMYND12, VPS52, HEXIM2, TRIM54, FSD2, FCHSD2, KLHL38, HOOK2* |
| Colon-sigmoid | *TCF4,* **POGZ**, *QRICH1, TSC1, YY1, SAE1* | *MEF2C, MEOX2, CCNDBP1, TRIM27, ODAM, INCA1, CDPF1, PAX8, FCHSD2, CCDC136, NFKBID, CDR2L, HOOK2, CDR2, NFYC, GOLGA2, VPS52, BICD2, TFIP11, BLZF1, CALCOCO2, PICK1, CCDC125, CADPS, MTUS2, YWHAE, AES, SSX2IP, BEGAIN, CEP70, MEF2A, TRIB3, FSD2, ZMYND12, TSGA10, HAP1,* |
| Colon-transverse | *HDAC4, SAE1,* **POGZ, ANKRD11**, *CLSTN3, EXOC5, KATNAL1, MEIS2, NSD1, SATB2, TCF4* | *SATB2, BCL2L15, KLC4, PTK6, A1CF, ABHD11, AGR2, NXPE2, AOC1* |
| Kidney | *CACNB2, TCF4, SAE1,* **POGZ**, *MEIS2, PHF12* | *MCCD1, AOC1, ATP6V0D2, CLCNKA, TMEM174, PAX8, A1CF* |
| Pituitary | *QRICH1, CIC, YY1, BAZ2B, USP7, USP15, HDAC4,* **ANKRD11**, *EP400, STXBP1, STX1A, MBD5, DPYSL2, WASF1,* **POGZ**, *DPYSL3, DAGLA, TCF4, MEIS2, DYNC1H1, NPAS2, EXOC5* | *RAB3IL1, BLOC1S6, ZNF696, ZNF440, PLN, SEC22A, TMEM254, KRT40, RMDN2, C1GALT1, NAPB, ZNF76, ZNF250, APOL2, STX12, VSTM4, BRD8, NEUROD4, AOC3, CENPP, CASC4, NINJ2, VAMP1, CLCNKA, TMEM41A, ZNF136, ABI3, STX7, STX2, VTI1B, STX4, CD81, EXOC5, APOL3, RHBDD2, ARL13B, VAMP4, STX3, BET1, ZNF12, TRAF3IP3, UBE2I, MAPK1, PGAP2, EBAG9, ZNF785, SMIM1, DEUP1, DDX49, STX10, ANKRD46, TRIM38, EFHC1, SERP1, C2orf82, CLEC1A, AIG1, CLN6, TXLNA, SERP2, VAMP5, JAGN1, TMEM120A, TSNARE1, MALL, TMEM199, C4orf3, ERG28, HMOX1, AGTRAP, SNAP47, STX16, USE1, CXCL16, BTN2A2, MAL, ZFPL1, TMEM222, FAM3C, BPIFA1, SPICE1, GOLGA2, CYB5B, GIMAP5, TMEM128, STX11, NKG7, STX6, LHFPL5, CMTM7, STX5, NRM, AQP3, VAMP3, BNIP1, LHX3, ETNK2, RNF4, TBX19, ZNF835, CDC37, ZNF441, KIFC3, STX8, TMEM100, ZNF707, TMEM60* |
| Lung | *PHF12,* | *ATP6V0D2* |
| Artery coronary | **ANKRD11** | *NOV* |
| Artery aorta | **ANKRD11** | *NOV* |
| Adipose subcutaneous | *ARNT2* | *SIM1* |
| Spleen | *SAE1,* **ANKRD11**, *TCF4, WASF1, CLSTN3,* **POGZ**, *STX1A, MEIS2, CAMK2B, TLK2, KHDRBS3, MARK1* | *POU2AF1, INPP5D, PAX5, DOCK2, NFAM1, GRB2, CCM2L, TCF23, TRAF3IP3, FOLR3, NKG7, RASAL3, ABI3, TCL1B* |
| Adrenal Gland | *TCF4, STX1A, EP400, QRICH1, TSC1,* **ANKRD11** | *MTFR1L, TMEM41A, NOV, CHCHD2, TRAPPC2L, RMDN2, FAM166B, ALAS1* |

Regulation of gene expression (96/200 genes in red, FDR = 4.21E-12, [Fig 6B]), Chromatin organization (44/200 in green, FDR = 1.44E-18, [Fig 6B]) and Histone modification (26/200, FDR = 2.08E-12) identified in the "*Biological Processes*", Chromatin binding (29/200, FDR = 3.00E-11), Transcription factor binding (24/200, FDR = 2.26E-06), DNA binding (53/

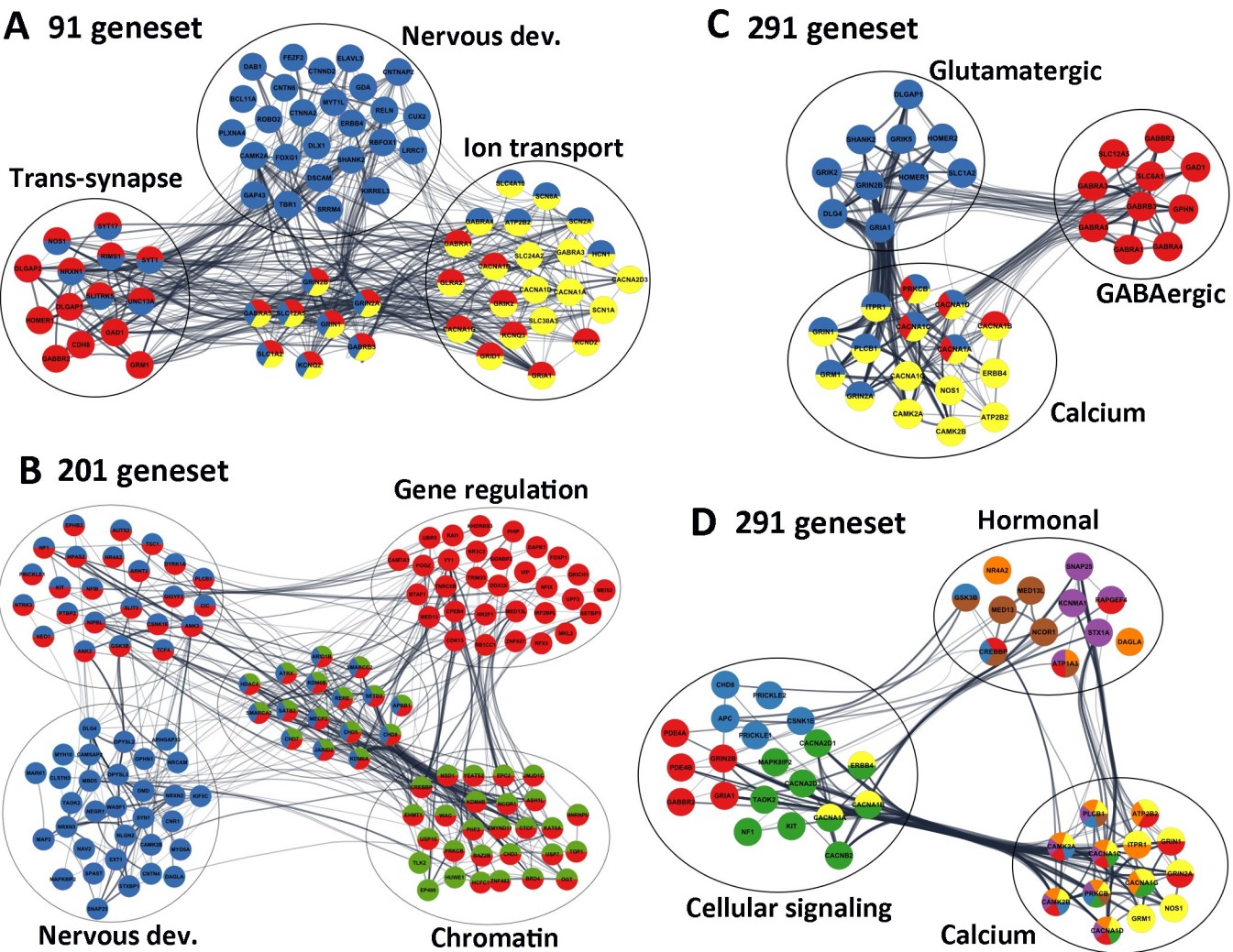

**Fig 6. Significant ASD pathways.** (A) Selected genes from 91 CNS geneset, highlighting that Nervous development (blue), Trans-synapses (red) and Ion transmembrane transportation (yellow) are the most enriched pathways. (B) Selected genes highlighted from the 200 CNS+PT geneset, demonstrating that the Nervous development (blue), the Chromatin organisation (green) and Gene regulation (red) were the among the most significant pathways. (C) Glutamatergic/GABAergic synapses, (D) Cell signalling and Hormonal secretion pathways from the combined 291 genes all linked to calcium signalling, suggesting that Calcium signalling is the most interconnected pathways linking the ASD signalling pathways. Edges represent combined gene score, node colours represent selected *GO* terms (A-B) and *KEGG* pathways (C,D) The colours for 3D correspond to the following; Yellow-Calcium Signalling, Green -MAPK Signalling, Red -cAMP signalling, Blue—Wnt Signalling, Brown—thyroid signalling, purple–Insulin Signalling, Orange–Aldosterone Synthesis and Secretion.

200, FDR = 6.72E-06) and Histone binding (12/200, FDR = 3.17E-05) in the "*Molecular Function*".

In addition, Circadian entrainment (5/200 genes, FDR = 0.034, *CACNA1C*, *CAMK2B*, *ITPR1*, *PLCB1*, *PRKCB*), WNT signalling (10/200, FDR = 3.1E-04, *APC*, *CAMK2B*, *CHD8*, *CREBBP*, *CSNK1E*, *GSK3B*, *PLCB1*, *PRICKLE1*, *PRICKLE2*, *PRKCB*), Thyroid hormone sig-nalling (8/200, FDR = 1.5E-03, *ATP1A3*, *CREBBP*, *GSK3B*, *MED13*, *MED13L*, *NCOR1*, *PLCB1*, *PRKCB*), Aldosterone synthesis and secretion (8/200, FDR = 5.0E-04, *ATP1A3*, *CACNA1C*, *CAMK2B*, *DAGLA*, *ITPR1*, *NR4A2*, *PLCB1*, *PRKCB*), Gastric acid secretion (5/200, FDR = 0.0179, *ATP1A3*, *CAMK2B*, *ITPR1*, *PLCB1*, *PRKCB*), Insulin secretion (9/200, FDR = 7.16E-05, *ATP1A3*, *CACNA1C*, *CAMK2B*, *KCNMA1*, *PLCB1*, *PRKCB*, *RAPGEF4*, *SNAP25*, *STX1A*), Salivary secretion (5/200, FDR = 0.0298, *ATP1A3*, *ITPR1*, *KCNMA1*,

**Table 3. Key Go terms of the CNS-specific and ubiquitous ASD genesets.**

| Term ID | Term description (Background Gene Count) | ASD genes | FDR | Matching proteins in the network (IDs) |
|---|---|---|---|---|
| **Key GO terms of the CNS-specific geneset (91 genes) centered on brain development, synapse and ion transport** | | | | |
| GO:0007399 (Biol. Proc.) | Nervous system development (2206) | 47 (CNS) | 1.39E-18 | ATP2B2, BCL11A, CAMK2A, CNTN6, CNTNAP2, CTNNA2, CTNND2, CUX2, DAB1, DLX1, DSCAM, ELAVL3, ERBB4, FEZF2, FOXG1, GABRA4, GABRA5, GABRB3, GAP43, GDA, GRIN1, GRIN2A, GRIN2B, HCN1, KCNQ2, KIRREL3, LRRC7, MYT1L, NOS1, NRXN1, PLXNA4, RBFOX1, RELN, RIMS1, ROBO2, SCN2A, SCN8A, SHANK2, SLC12A5, SLC1A2, SLC4A10, SLITRK5, SRRM4, SYT1, SYT17, TBR1, UNC13A |
| GO:0043005 (Cell. Comp.) | Neuron projection (1142) | 44 (CNS) | 9.54E-28 | ANKS1B, CACNA1B, CADM2, CAMK2A, CDH8, CNKSR2, CNTNAP2, CTNNA2, CTNND2, DAB1, DSCAM, FRMPD4, GABBR2, GABRA5, GAD1, GAP43, GRIA1, GRIK2, GRIN1, GRIN2A, GRIN2B, GRM1, HCN1, HOMER1, KCND2, KCNQ2, KCNQ3, KIRREL3, LRRC4, LRRC7, NOS1, NRXN1, RELN, ROBO2, SCN1A, SCN2A, SCN8A, SHANK2, SLC12A5, SLC1A2, SLC30A3, SLC4A10, SYT1, UNC13A |
| GO:0099537 (Biol. Proc.) | Trans-synaptic signalling 408) | 31 (CNS) | 2.41E-25 | CACNA1B, CACNA1G, CDH8, DLGAP1, DLGAP2, GABBR2, GABRA1, GABRA5, GABRB3, GAD1, GLRA2, GRIA1, GRID1, GRIK2, GRIN1, GRIN2A, GRIN2B, GRM1, HOMER1, KCND2, KCNQ2, KCNQ3, NOS1, NRXN1, RIMS1, SLC12A5, SLC1A2, SLITRK5, SYT1, SYT17, UNC13A |
| GO:0045202 (Cell. Comp.) | Synapse (107) | 42 (CNS) | 4.06E-30 | ANKS1B, ATP2B2, CACNA1B, CADM2, CAMK2A, CDH8, CNKSR2, CTNNA2, DAB1, DLGAP1, DLGAP2, DLGAP3, DSCAM, FRMPD4, GABBR2, GABRA1, GABRA3, GABRA4, GABRA5, GABRB3, GAD1, GAP43, GLRA2, GRIA1, GRID1, GRIK2, GRIN1, GRIN2A, GRIN2B, GRM1, HOMER1, KCND2, LRRC4, LRRC7, NOS1, NRXN1, RIMS1, SHANK2, SLC30A3, SYT1, SYT17, UNC13A |
| GO:0030594 (Mol. Funct.) | Neurotransmitter receptor activity (849) | 13 (CNS) | 3.11E-13 | GABBR2, GABRA1, GABRA3, GABRA4, GABRB3, GLRA2, GRIA1, GRID1, GRIK2, GRIN1, GRIN2A, GRIN2B, GRM1 |
| GO:0008066 (Mol. Funct.) | Glutamate receptor activity (27) | 7 (CNS) | 2.73E-09 | GRIA1, GRID1, GRIK2, GRIN1, GRIN2A, GRIN2B, GRM1 |
| GO:0016917 (Mol. Funct.) | GABA receptor activity (22) | 6 (CNS) | 3.86E-08 | GABBR2, GABRA1, GABRA3, GABRA4, GABRA5, GABRB3 |
| GO:0034220 (Biol. Proc.) | Ion transmembrane transport (995) | 30 (CNS) | 1.55E-14 | ATP2B2, CACNA1A, CACNA1B, CACNA1D, CACNA1G, CACNA2D3, GABRA1, GABRA3, GABRA4, GABRA5, GABRB3, GLRA2, GRIA1, GRID1, GRIK2, GRIN1, GRIN2A, GRIN2B, HCN1, KCND2, KCNQ2, KCNQ3, SCN1A, SCN2A, SCN8A, SLC12A5, SLC1A2, SLC24A2, SLC30A3, SLC4A10 |
| GO:0034702 (Cell. Comp.) | Ion channel complex (278) | 24 (CNS) | 7.59E-22 | CACNA1A, CACNA1B, CACNA1D, CACNA1G, CNTNAP2, DPP10, GABRA1, GABRA3, GABRA4, GABRA5, GABRB3, GLRA2, GRIA1, GRIK2, GRIN1, GRIN2A, GRIN2B, HCN1, KCND2, KCNQ2, KCNQ3, SCN1A, SCN2A, SCN8A |
| GO:0022839 (Mol. Funct.) | Ion gated channel activity (329) | 24 (CNS) | 4.34E-19 | CACNA1A, CACNA1B, CACNA1D, CACNA1G, CACNA2D3, GABRA1, GABRA3, GABRA4, GABRA5, GABRB3, GLRA2, GRIA1, GRID1, GRIK2, GRIN1, GRIN2A, GRIN2B, HCN1, KCND2, KCNQ2, KCNQ3, SCN1A, SCN2A, SCN8A |
| GO:0005262 (Mol. Funct.) | Calcium channel activity (114) | 9 (CNS) | 7.18E-08 | CACNA1A, CACNA1B, CACNA1D, CACNA1G, CACNA2D3, GRIN1, GRIN2A, RIN2B, SLC24A2 |
| **Key GO terms of the CNS+PT geneset (200 genes) clustered on brain development, synapse, and gene regulation** | | | | |
| GO:0007399 (Biol. Proc.) | Nervous system development (2206) | 72 (CNS+PT) | 7.50E-17 | ANK2, ANK3, APBB1, ARHGAP33, ARID1B, ARNT2, ATRX, AUTS2, CAMK2B, CAMSAP2, CHD5, CHD7, CHD8, CIC, CLSTN3, CNR1, CNTN4, CSNK1E, DAGLA, DLG4, DMD, DPYSL2, DPYSL3, DYRK1A, EPHB2, EXT1, GIGYF2, GSK3B, HDAC4, JARID2, KDM6A, KDM6B, KIF5C, KIT, MAP2, APK8IP2, MARK1, MBD5, MECP2, MYH10, MYO5A, NAV2, NEGR1, NEO1, NF1, NFIB, NIPBL, NLGN2, NPAS2, NR4A2, NRCAM, NRXN2, NRXN3, NTRK3, OPHN1, PLCB1, PRICKLE1, PTBP2, RERE, SATB2, SETD2, SLIT3, SMARCA2, SMARCC2, SNAP25, SPAST, STXBP1, SYN1, TAOK2, TCF4, TSC1, WASF1 |
| GO:0050804 (Biol. Proc.) | Modulation of chemical synaptic transmission (316) | 20 (CNS+PT) | 1.74E-08 | CAMK2B, CLSTN2, CLSTN3, CNR1, CNTN4, DLG4, EPHB2, GRIK5, KCNB1, KIT, MAPK8IP2, MECP2, NF1, NLGN2, OPHN1, RIMS3, SNAP25, STX1A, STXBP1, SYN1 |
| GO:0099537 (Biol. Proc.) | Trans-synaptic signalling (408) | 22 (CNS+PT) | 3.16E-08 | ARID1B, CACNB2, CASK, CLSTN3, CNR1, DAGLA, DLG4, EPHB2, GRIK5, GSK3B, MAPK8IP2, MECP2, MYO5A, NF1, NRXN2, RIMS3, SLC6A1, SNAP25, STX1A, STXBP1, SYN1, SYNJ1 |

*(Continued)*

**Table 3.** (Continued)

| Term ID | Term description (Background Gene Count) | ASD genes | FDR | Matching proteins in the network (IDs) |
|---|---|---|---|---|
| GO:0045202 (Cell. Comp.) | Synapse (849) | 45 (CNS +PT) | 3.25E-17 | *ANK2, ANK3, APBB1, ARHGAP32, ARHGAP33, ATP1A3, CACNA1C, CADM1, CAMK2B, CASK, CLSTN2, CLSTN3, CNR1, CPEB4, DAGLA, DLG4, DMD, DMXL2, EPHB2, GPHN, GRIK5, GSK3B, HDAC4, ITPR1, KCNB1, MAP2, MAPK8IP2, MECP2, MYH10, NF1, NLGN2, NRCAM, NRXN2, OPHN1, PDE4B, PSD3, RIMS3, SNAP25, STX1A, STXBP1, STXBP5, SYN1, SYNE1, SYNJ1, WASF1* |
| GO:0005634 (Cell. Comp.) | Nucleus (6892) | 128 (CNS +PT) | 1.73E-14 | *ANKRD11, APBB1, APC, ARHGAP32, ARID1B, ARNT2, ASH1L, ATRX, AUTS2, BAZ2B, BRD4, BTAF1, CAMK2B, CAMTA1, CASK, CDK13, CHD3, CHD5, CHD7, CHD8, CIC, CMIP, CPEB4, CREBBP, CSNK1E, CTCF, DCUN1D1, DDX3X, DMD, DOCK4, DST, DYRK1A, EHMT1, EP400, EPC2, EPHB2, FBXO11, FOXP1, GGNBP2, GRIK5, GSK3B, HCFC1, HDAC4, HERC2, HNRNPU, HUWE1, INTS6, IRF2BPL, ITPR1, JARID2, JMJD1C, KAT6A, KATNAL1, KDM4B, KDM6A, KDM6B, KHDRBS3, MAP2, MBD5, MECP2, MED13, MED13L, MEIS2, MKL2, NAV2, NCOR1, NEO1, NF1, NFIB, NFIX, NIPBL, NPAS2, NR2F1, NR3C2, NR4A2, NSD1, OCRL, OFD1, OGT, PDE4A, PHF2, PHIP, PLCB1, POGZ, PRICKLE1, PRICKLE2, PRKCB, PRPF39, PTBP2, QRICH1, RAI1, RB1CC1, RBM27, RERE, RFX3, RNF38, SAE1, SATB2, SETBP1, SETD2, SMARCA2, SMARCC2, SPAST, SRSF11, STX1A, STXBP1, SYN1, SYNE1, TAOK2, TCF4, TLK2, TOP1, TRIM33, TRIP12, TSC1, UBN2, UBR5, UPF2, USP15, USP7, WAC, WDFY3, WDR26, YEATS2, YY1, ZMYND11, ZNF462, ZNF827* |
| GO:0006325 (Biol. Proc.) | Chromatin organization (683) | 44 (CNS +PT) | 1.44E-18 | *APBB1, ARID1B, ASH1L, ATRX, BAZ2B, BRD4, CHD3, CHD5, CHD7, CHD8, CREBBP, CTCF, EHMT1, EP400, EPC2, HCFC1, HDAC4, HNRNPU, HUWE1, JARID2, JMJD1C, KAT6A, KDM4B, KDM6A, KDM6B, MECP2, NCOR1, NSD1, OGT, PHF2, PRKCB, RERE, SATB2, SETD2, SMARCA2, SMARCC2, TLK2, TOP1, USP15, USP7, WAC, YEATS2, ZMYND11, ZNF462* |
| GO:0003682 (Mol. Func.) | Chromatin binding (501) | 29 (CNS +PT) | 2.62E-11 | *APBB1, ASH1L, ATRX, AUTS2, BRD4, CHD7, CHD8, CIC, CREBBP, CTCF, EP400, HCFC1, HDAC4, HNRNPU, JARID2, JMJD1C, KDM6A, KDM6B, MBD5, MECP2, NIPBL, NSD1, PRKCB, RERE, SATB2, SMARCA2, TOP1, WAC, ZMYND11* |
| GO:0016570 (Biol. Proc.) | Histone modification (347) | 26 (CNS +PT) | 2.08E-12 | *APBB1, ASH1L, CHD3, CHD5, CREBBP, EHMT1, EP400, EPC2, HCFC1, HDAC4, HUWE1, JMJD1C, KAT6A, KDM4B, KDM6A, KDM6B, MECP2, NSD1, OGT, PHF2, PRKCB, SETD2, USP15, USP7, WAC, YEATS2* |
| GO:0042393 (Mol. Func.) | Histone binding (188) | 12 | 2.99E-05 | *APBB1, ATRX, BRD4, CHD5, CHD8, PHF2, PHIP, PRKCB, SMARCA2, USP15, YEATS2, ZMYND11* |
| GO:0008134 (Mol. Func.) | Transcription factor binding (610) | 24 | 2.05E-06 | *APBB1, ARNT2, CDK13, CREBBP, DDX3X, FOXP1, GSK3B, HCFC1, HDAC4, HNRNPU, JARID2, JMJD1C, KAT6A, MECP2, MED13, MEIS2, NCOR1, NR4A2, NSD1, PRKCB, TCF4, TRIP12, USP7, YEATS2* |
| GO:0010468 (Biol. Proc.) | Regulation of gene expression (4533) | 96 (CNS +PT) | 4.21E-12 | *ANK2, ANK3, APBB1, ARID1B, ARNT2, ASH1L, ATRX, AUTS2, BAZ2B, BRD4, BTAF1, CAMTA1, CDK13, CHD3, CHD5, CHD7, CHD8, CIC, CPEB4, CREBBP, CSNK1E, CTCF, DAPK1, DDX3X, DYRK1A, EHMT1, EPC2, EPHB2, FOXP1, GGNBP2, GIGYF2, GSK3B, HCFC1, HDAC4, HNRNPU, IRF2BPL, JARID2, JMJD1C, KAT6A, KDM4B, KDM6A, KDM6B, KHDRBS3, KIT, MECP2, MED13, MED13L, MEIS2, MKL2, NCOR1, NEO1, NF1, NFIB, NFIX, NIPBL, NPAS2, NR2F1, NR3C2, NR4A2, NSD1, NTRK3, OGT, PHF2, PHIP, PLCB1, POGZ, PRICKLE1, PRKCB, PTBP2, QRICH1, RAI1, RB1CC1, RERE, RFX3, SATB2, SETBP1, SETD2, SLIT3, SMARCA2, SMARCC2, TCF4, TNRC6B, TOP1, TRIM33, TSC1, UBR5, UPF2, USP15, USP7, VIP, WAC, YEATS2, YY1, ZMYND11, ZNF462, ZNF827* |
| GO:0003677 (Mol. Func.) | DNA binding (2457) | 53 | 5.62E-06 | *ARID1B, ARNT2, ASH1L, ATRX, BAZ2B, BTAF1, CAMTA1, CDK13, CHD3, CHD5, CHD7, CHD8, CIC, CREBBP, CTCF, DDX3X, EP400, FOXP1, HDAC4, HNRNPU, HUWE1, JARID2, JMJD1C, KAT6A, KDM6A, KDM6B, MECP2, MEIS2, NCOR1, NFIB, NFIX, NPAS2, NR2F1, NR3C2, NR4A2, NSD1, POGZ, QRICH1, RAI1, RERE, RFX3, SATB2, SETBP1, SMARCA2, SMARCC2, TCF4, TOP1, TRIM33, UPF2, YY1, ZMYND11, ZNF462, ZNF827* |

(*Continued*)

**Table 3.** (Continued)

| Term ID | Term description (Background Gene Count) | ASD genes | FDR | Matching proteins in the network (IDs) |
|---|---|---|---|---|
| GO:0043167 (Mol. Func.) | Ion binding (6066) | 106 (CNS +PT) | 1.42E-08 | AGAP1, ARHGAP32, ARHGAP33, ASAP2, ASH1L, ATP1A3, ATP8A1, ATRX, BAZ2B, BTAF1, CACNA1C, CACNA2D1, CAMK2B, CASK, CDC42BPB, CDK13, CHD3, CHD5, CHD7, CHD8, CLSTN2, CLSTN3, CPEB4, CREBBP, CSNK1E, CTCF, DAGLA, DAPK1, DDX3X, DMD, DPYSL3, DST, DYNC1H1, DYRK1A, EHMT1, EP400, EPHB2, EXT1, FBXO11, FGD1, FOXP1, GPHN, GSK3B, HDAC4, HERC2, HNRNPU, IRF2BPL, ITPR1, JMJD1C, KAT6A, KATNAL1, KCND3, KCNMA1, KDM4B, KDM6A, KDM6B, KIF21B, KIF5C, KIT, MARK1, MYH10, MYO5A, NAV2, NF1, NPAS2, NR2F1, NR3C2, NR4A2, NRXN2, NRXN3, NSD1, NTRK3, OGT, OPHN1, PDE4A, PDE4B, PHF2, PHF3, PLCB1, POGZ, PRICKLE1, PRICKLE2, PRKCB, RAI1, RAPGEF4, RBM27, RERE, RNF38, SETD2, SLC6A1, SLIT3, SMARCA2, SPAST, SYN1, TAOK2, TLK2, TOP1, TRIM33, TRIO, UBR5, WDFY3, YTHDC2, YY1, ZMYND11, ZNF462, ZNF827 |

*PLCB1*, *PRKCB*) and Pancreatic secretion (5/200, FDR = 0.0343, *ATP1A3*, *ITPR1*, *KCNMA1*, *PLCB1*, *PRKCB*) were also detected as significant KEGG pathways in the CNS+PT ASD geneset.

In consistency with this, STRING analysis of the combined 291 ASD genes gave rise to 47 "*KEGG* pathways" (S18 Table), 677 "*Biological Processes*" (S19 Table), 149 "*Cellular Components*" (S20 Table) and 177 "*Molecular Function*" (S21 Table). The top enriched pathways of the combined ASD risk factors included Nervous system development (119/291 genes, FDR = 8.85E-34), Synapse (87/291, 1.9E-43), Trans-synaptic signalling (53/291, 1.04E-28), Ion channel complex (36/291, 2.55E-20), Ion-gated channel activity (33/291, 1.2E-14), Regulation of ion transport (44/291, 4.79E-15) and Neurotransmitter receptor activity (14/291, 8.86E-08), Chromatin organisation (44/291, 9.4E-14), Chromosome (36/291, 4.80E-06) and Chromatin binding (31/291, 3.48E-09) and Positive regulation of gene expression (66/291, 8.15E-10). These data suggest that the CNS+PT geneset of ASD candidate genes may influence not only the core symptoms via CNS development and synaptic function, but also the comorbidities through dysregulated gene expression and hormonal signalling in the peripheral organs.

## E/I balance, and calcium signalling are central to ASD

KEGG analyses generated 26 pathways for the CNS (S13 Table) geneset and 31 pathways for the CNS+PT geneset (S17 Table). The top pathways from the CNS geneset corresponded to Glutamatergic synapse (12/91 genes, 2.46E-11), GABAergic synapse (11/91, 3.44E-11), Neuroactive ligand-receptor interaction (14/91, 1.42E-09), Retrograde endocannabinoid signalling (11/91, 3.77E-09), Calcium signalling pathway (11/91, 1.80E-08) and MAPK signalling (6/91, 0.0105). The CNS+PT ASD geneset was also enriched for Secretion, Thyroid function, and Cellular signalling. Interestingly, 10 pathways from both ASD genesets appeared to overlap, which included Glutamatergic, Dopaminergic, Cholinergic synapses, Circadian entrainment, cAMP and Retrograde endocannabinoid signalling (Table 3). Calcium channels, Glutamatergic and GABAergic receptors appeared to link to most of the pathways.

To further investigate the interconnectivity among the KEGG pathways (S18 Table), we constructed an interaction matrix with the combined ASD geneset KEGG pathways (S22 Table), similar to a previous analysis [30]. The calcium signalling (27/47), MAPK Signalling (12/47) and cAMP signalling (6/47) were identified as highly interconnected signalling pathways. Pre-synaptically, the genes in calcium signalling also appeared frequently in the other pathways, particularly the genes encoding the pore-forming subunit of voltage-gated calcium channel (*CACNA1A*, *CACNA1B*, *CACNA1C*, *CACNA1D*), the I3P receptor *IPTR1*, *PLCB* and the calmodulin proteins (*CAMK2A*, *CAMK2B*) are connected to the neurotransmitter releases

**Table 4. Genes dysregulated in ASD expression studies that overlapped with our geneset.**

| Study | Tissue type | genes overlap | up | down |
|---|---|---|---|---|
| Gupta [66] | Cortical post-mortem | 22 | 18 | 4 |
| Parikshak [67] | Cortical post-mortem | 24 | 4 | 20 |
| Voineagu [56] | Cortical post-mortem | 48 | 39 | 9 |
| Walker (2013) [49] | colon | 2 | 0 | 2 |
| Walker (2016) [46] | blood and colon | 70 | 5\|37** | 29\|6** |
| Mariani [50] | Organoids | 89 | 86* | 3* |
| Gresi Olivera [54] | IPSC neurons | 2 | 2 | 0 |
| DeRosa [53] | IPSC neurons | 15 | 4* | 11* |
| Velmeshev [68] | Cortical tissue post-mortem | 38 | 32 | 6 |
| Herrero [69] | Amygdala post-mortem | 9 | 2 | 7 |
| Chien [48] | Lymphblastoid cell lines | 9 | 2 | 7 |
| Ginsberg [70] | Cortical tissue | 2 | 2 | 0 |
| Garbett [45] | Superior Temporal Gyrus | 2 | 0 | 2 |
| Breen [55] | IPSC neurons and NPCs | 10 | 9 | 1 |
| Wang (2015) [64] | IPSC neurons and NPCs | 84 | 63 | 21 |
| Wang (2017) [52] | Organoids | 24 | 17 | 7 |
| Pramparo [47] | Leukocytes | 15 | 0 | 12 |

*denotes genes which have simultaneous up/down expression

** denote genes for blood and colon samples respectively.

(Table 4). Post-synaptically, Glutamatergic synapse (8/47) had the largest number of interactions with other pathways, which was followed by Dopaminergic synapse (7/47). The NDMAR (*GRIN2A*, *GRIN2B*, *GRIN1*) and AMPA (*GRIA1*) receptors were also the highly interconnected notes in neural and synaptic, calcium and cAMP signalling pathways. In summary, this overlap analyses demonstrated that the E/I balance, and calcium signalling are the most significant pathways linking to the ASD core symptoms, and transcriptional regulation to the ASD comorbidities.

## Enriched epilepsy/seizures in the CNS geneset and congenital abnormalities/developmental delay in CNS+PT geneset

The results from WebGestalt (Fig 7, Supplementary links 1 and 2) showed that epilepsy and seizures were enriched in 19 of the top 50 phenotype terms and movement disorders in 9 of the 50 terms in the CNS geneset. In the CNS+PT set, there was enrichment for behaviour issues (9/50) such as self-injurious (FDR = 1.04E-07), aggressive behaviour (1.04E-11), and congenital abnormalities of the face (25/50 items).

## High proportion of the ASD genes were dysregulated in ASD

We compared the ASD geneset with DEGs between ASD and controls in the literature and found that 201 of the 292 genes in our ASD geneset were dysregulated, at least once across multiple ASD gene expression studies (Table 4, S23 Table). Recurrent DEGs such as RELN, FOXP1, GAD1, NRXN1, FOXG1 and CAMK2A were present in multiple studies (S23 Table). These data suggest that not only mutations but also dysregulated expression of the ASD risk genes can be linked to the development of ASD.

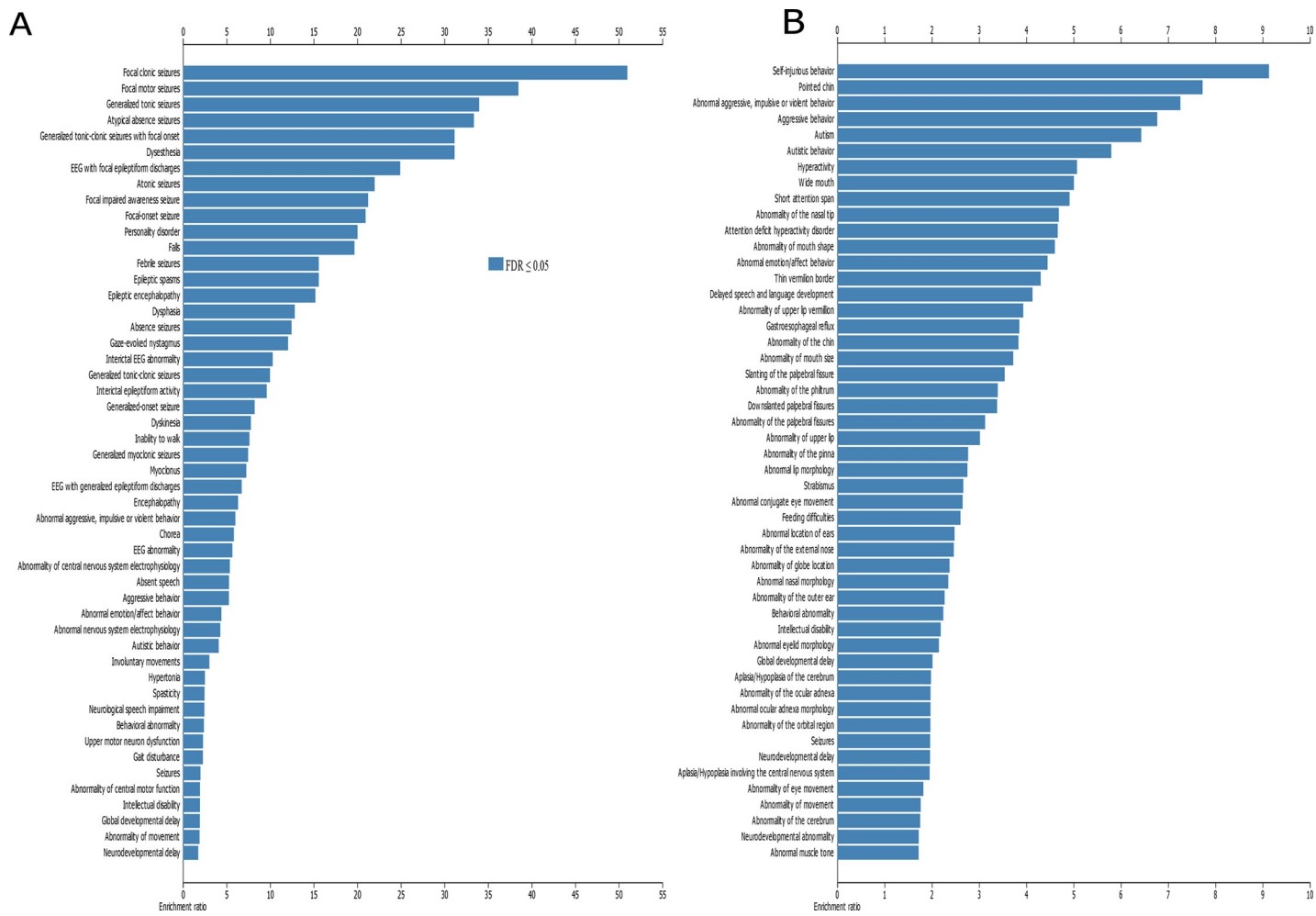

**Fig 7. Co-Morbid phenotypes associated with ASD.** (A) Top 50 terms enriched in CNS geneset, (B) top 50 terms enriched in CNS+PT geneset. X-axis denotes ratio of enrichment. Dark blue are terms below FDR < 0.05.

## ASD genes enriched in cortex development and sex-bias brain expression

We compared ASD shortlist with CSEA dataset and found that 280 of the 292 ASD genes were mapped to CSEA, with an enrichment in the cortex across most timepoints (S24 Table), with most significant enrichment in the early-mid fetal stage of brain development at pSI 0.05 (p-value = 5.427e-16, FDR = 3.256e-14) pSI 0.01 (p-value = 5.560e-07 FDR = 3.336e-05) and pSI 0.001 p-value = 8.912e-04, FDR = 0.053) (Fig 8D). In addition, there was also enrichment of genes in the striatum at the early-mid fetal stage (p-value = 1.872e-08, FDR = 2.808e-07), in the cerebellum at childhood (p-value = 7.576e-07, FDR = 7.576e-06) and adolescence (p-value = 9.970e-04, FDR = 0.005), in the thalamus in early fetal (p-value = 0.010, 0.043) and neonatal period (p-value 4.333e-04, FDR = 0.003), and in the amygdala from mid-early (p-value = 0.009, FDR = 0.042) to mid-late (p-value = 0.017, FDR = 0.064) fetal stages. These data suggest that the ASD geneset plays essential role in brain development, especially corticogenesis.

To investigate if the ASD genes are sex-related, we compared the ASD geneset with genes which were known to have sex-biased expression in prenatal male and female (S25 Table). Significant correlations were found with genes bias for female cerebellar cortex, dorsolateral

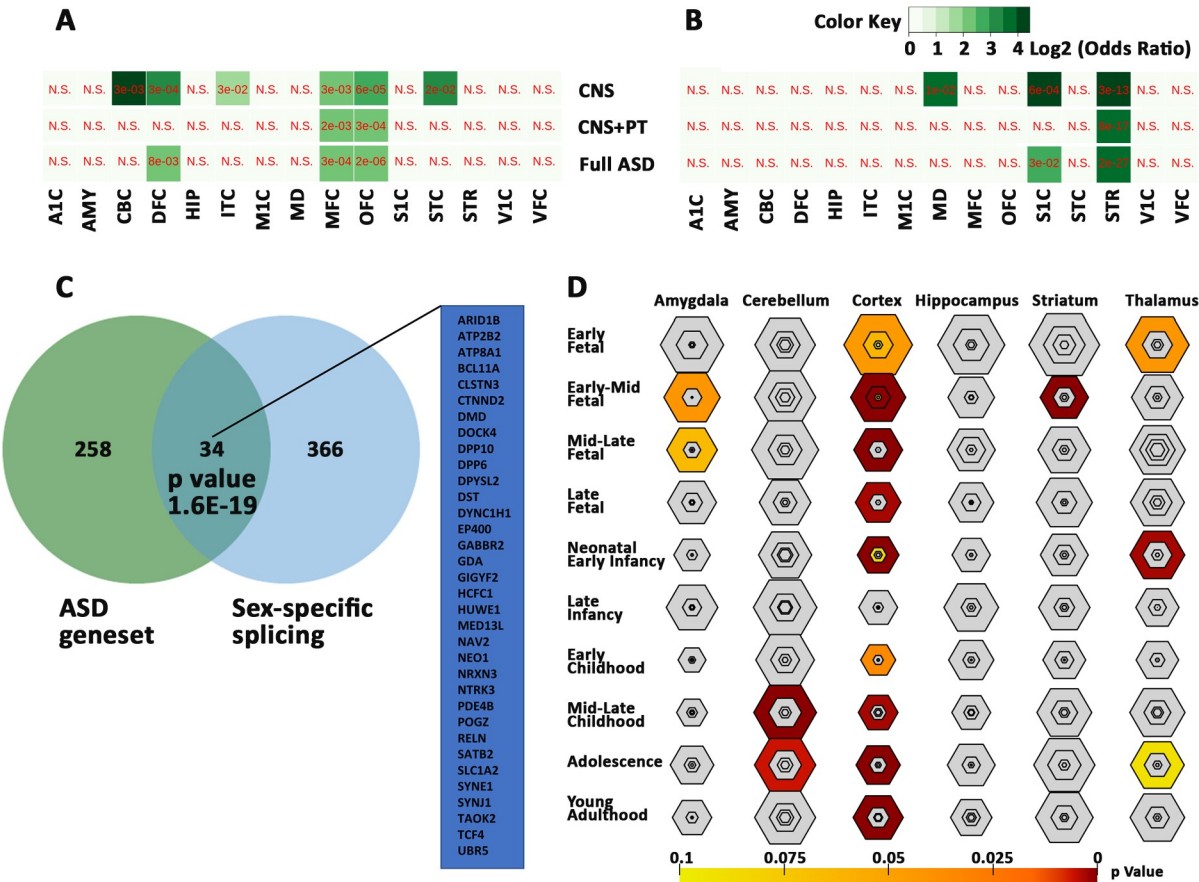

**Fig 8. Sex-biased expression of the ASD genesets.** Biased expression of the ASD genesets in female (A) and male (B) brain regions. Values are expressed as FDR in red and Log2 odds ratio in green with gradients. N.S for not significant. (C) Overlap of 34 ASD genes with sex-biased splicing genes. (D) Bullseye plot showing enrichment of ASD geneset in brain regions with most significant correlation in cerebral cortex throughout brain development. A1C -primary auditory cortex, AMY—amygdaloid complex, CBC—cerebellar cortex, DFC -dorsolateral prefrontal cortex, HIP—hippocampus, IPC—posterior inferior parietal cortex, ITC—inferolateral temporal cortex, M1C-primary motor cortex, MD—mediodorsal nucleus of thalamus, MFC—medial prefrontal cortex, OFC -orbital frontal cortex, S1C - primary somatosensory cortex, STC—posterior(caudal) superior temporal cortex, STR -striatum, VFC—ventrolateral prefrontal cortex, and V1C - primary visual cortex.

prefrontal cortex, medial prefrontal cortex, orbital frontal cortex, inferolateral temporal cortex and caudal superior temporal cortex (Fig 8A), and with genes bias for male primary somato-sensory cortex, mediodorsal nucleus of thalamus and striatum (Fig 8B). In addition, 34 of the 292 genes were found to have sex-biased gene-splicing in at least one brain region (Fig 8C, S25 Table). These genes are likely to contribute to sex-bias occurrence of the ASD.

## ASD genes present in other conditions

We next compared the ASD genes with other neuropsychiatric conditions, including schizophrenia, bipolar and major depression, and peripheral conditions such as arrythmia, inflammatory bowel disease and type 1 and type 2 diabetes (Fig 9, S25 Table), and identified 94 genes which were identified in Pyschgenet database (Fig 9A), and involved in synaptic transmission. The remaining ASD-unique genes were enriched for chromatin organization and gene expression. We also found significant overlaps of the ASD genes with factors associated with psychiatric and peripheral conditions (Fig 9B), including a large overlap with type 2 diabetes (134/

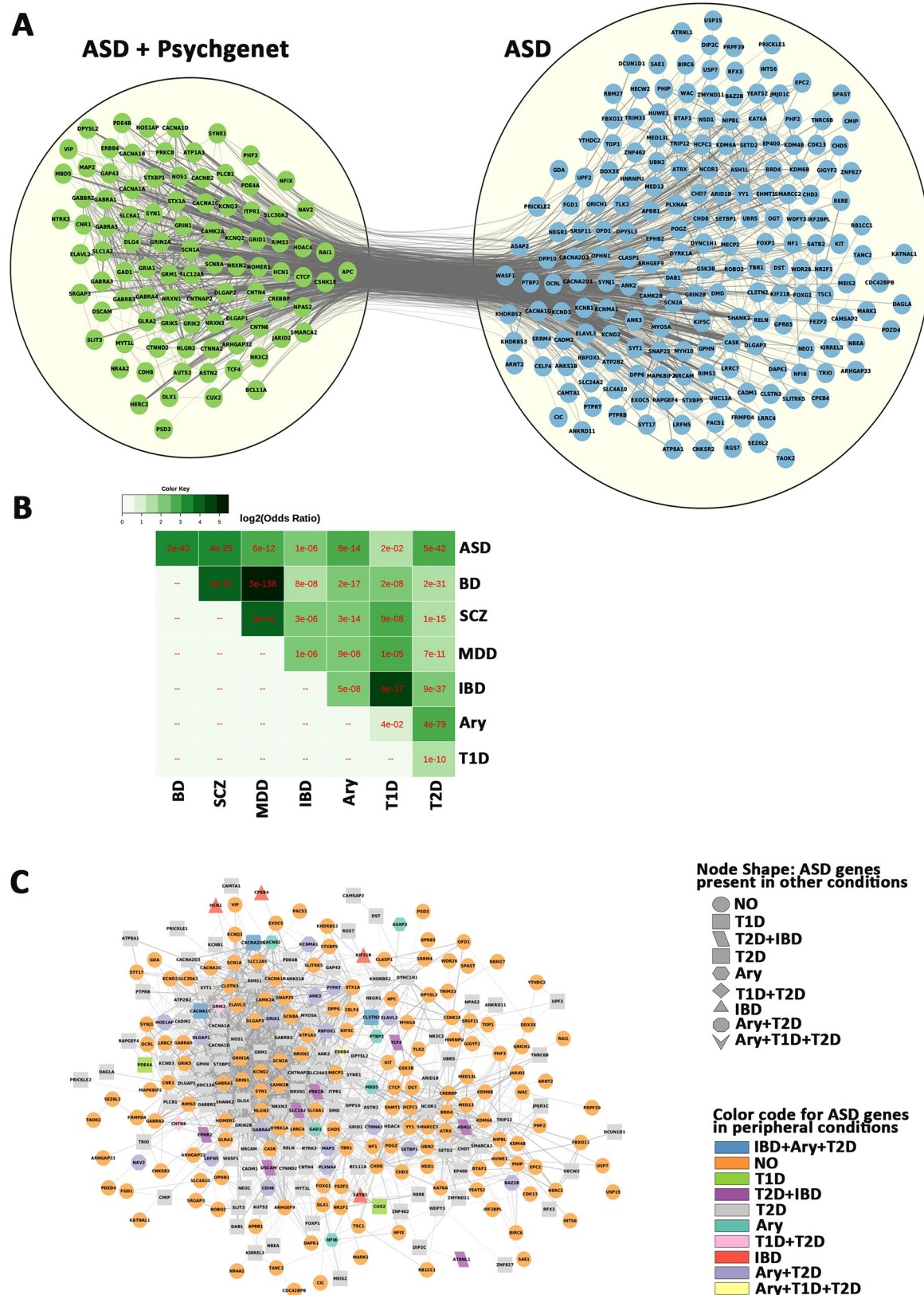

**Fig 9. Overlap of ASD genes with other conditions.** A) Comparison of the ASD geneset with Pyschgenet (SCZ, BP, MDD) defines a network of 94 overlapping genes (green) and 198 ASD-unique genes (blue). B) matrix of overlaps between ASD/Psychiatric (SCZ, BP, MDD) and peripheral (IBD, ARY. T1D and T2D) conditions. C) Network of ASD genes and overlaps with peripheral (IBD, ARY. T1D and T2D) conditions, with orange circle for non-overlapping genes. IBD—Inflammatory bowel disease, Ary—Arrythmia, T1D - type 1 diabetes, T2D - type 2 diabetes. SCZ–schizophrenia, BP–bipolar disorder, MDD–major depressive disorder.

292, FDR = 3e-42). These data suggest a common disturbance in neuronal communication with CNS other neuropsychiatric disorders and gene dysregulation with peripheral conditions.

## Discussion

It is becoming apparent that ASD genes could influence other organ systems. This is reflected by the many co-morbidities occurring outside the CNS such gastrointestinal issues, metabolic disorders, auto-immune disorders, tuberous sclerosis, attention-deficit hyperactivity disorder, and sensory problems associated motor problems. However, little attention has been given to related organs of major comorbidities. Here we have identified 319 overlapping ASD candidates among the four independent scoring systems and the SFARI database [28, 31–34]. We also introduced gene expression using the GTEx database [39, 71], which consists of mRNA data of 53 human tissues from approximately 1000 individuals at the age of 21–70. This resulted in a shortlist of 292 common ASD candidate genes with mRNA expression at TPM ≥3 transcripts. This also categorized the ASD factors into 2 genesets, the CNS-specific geneset of 91 genes (with a TMP<3 in PT) and the CNS+PT geneset of 201 genes. This was validated at the protein level across these tissues in the human protein atlas (HPA) dataset and Huri, showing that that ASD genes are not only expressed in other organs outside the brain, but also appear to interact with other proteins in tissue-specific networks.

STRING analyses show that the CNS geneset is enriched for nervous development, glutamatergic/GABAergic synapses, and calcium signalling. Phenotype analysis also showed high enrichment for epilepsy and seizures. Both results support the hypothesis that disruption of E/I balance during CNS development as a major feature of ASD [72], which are related to CNS co-morbidities such as epilepsy occurring in 30% of ASD at severe end of the spectrum.

The expression of 201 ASD candidate genes in CNS+PT suggest that ASD genes may influence not only the CNS but also peripheral systems in the body. This geneset is enriched for nervous development and synapse, as well as for chromatin organisation and gene regulation, which are consistent with previous reports from exome sequencing studies [12, 14, 16, 17, 29]. Therefore, the genes involved in chromatin organisation and gene regulation could have an influence in peripheral co-morbidities. Many of the genes in our ASD geneset also show dysregulated expression in multiple studies (Table 4) pertaining to cortical tissue and iPSC derived models, and in the few studies carried on gene expression in the gastrointestinal tract of ASD.

Strong candidate genes such as *CHD8*, *POGZ* and *DYRK1A* were previously reported to be associated with not only autism but also gastrointestinal issues, facial dysmorpisms, visual and feeding problems [73–76]. Indeed, facial dysmorphism is also a recurrent phenotype that emerges among various subgroups of autistics with known genetic mutations [77–79]. Some enrichment of ASD genes reported to overlap with heart development (S10-21) and congenital heart deformation [80, 81], and a high rate of ASD diagnosis was reported among children with congenital heart defects [82]. *POGZ* and *ANKRD11* are also present in many tissue interaction networks (Table 2) which are involved in neural proliferation. The *POGZ* is a zinc finger protein interacting with the transcription factor SP1 [83], and *ANKRD11* is a chromatin regulator, modulating histone acetylation and inhibiting ligand-dependent activation of transcription [84]. *ANKRD11* mutations have been associated with diseases with distinctive craniofacial

features, short stature, skeletal anomalies, global developmental delay, seizures and intellectual disability [85].

The strong enrichment for neuronal processes and functions in tissues beyond the brain is of curious interest, but researchers are starting to explore other aspects of ASD that could have an impact on other organs such as the heart via the sympathetic and parasympathetic nervous systems [86], and the gastrointestinal tract via the enteric nervous system [87]. In fact there is growing evidence that heart rate is affected among Autistics [88–91], along with evidence that ASD genes could be involved in aspects of gastrointestinal development and function [74, 92–97]. There is a potential for more work on how the parts of the brain control heart rate and gastrointestinal, how they are altered in autism, or even if reported autonomic issues in co-morbidities such gastroesophageal reflux [98, 99] hold true in the autistic population as well. The expression of proteins such as *STX1A*, *SNAP25* and *FOXP1* expressed in endocrine tissues could be of interest in ASD, given how knockouts in these genes can impact the development and function of certain parts of the endocrine system [100–102] and how genes involved in neurotransmission could also be involved in secretion of hormones [103].

Another unaddressed question is if the peripheral nervous system and peripheral organs are affected by mutations in addition to the CNS. Amongst the results we found enrichment for neuromuscular and cardiac function in STRING analysis (S9-20). Some animal models such as *FOXP1*, *SHANK3*, *NOS1* and *CHD8* [74, 92, 94, 96, 97] have been developed for functional analysis of ASD candidate genes in the gastrointestinal system, which indicates that ASD genes may play an important role in this organ [95], and yet most research have been mainly focused on the brain in both human and animal models. A greater utilisation of the animal models to explore other systems such as the cardiac and gastrointestinal systems would be welcome.

In fact, 88/292 genes in our ASD geneset already have existing genetic mouse models, as well as rescue models for 28/292 genes according to the SFARI database (July 2020) at the time of writing. They are helpful in understanding function of these genes, and may also assist drug development to remediate related pathways, not just in the brain, but also throughout the peripheral nervous system that connects to co-morbidity organs, and even in the peripheral organs themselves.

The interconnectivity analyses from the current study reveal calcium, MAPK and glutamatergic signalling as three highest interconnected pathways, all are also involved with each other based on the interaction matrix. This is in line with a previous publication that ASD factors are converged upon MAPK and calcium signalling [30]. It is worth to note that MAPK signalling is also interlinked with calcium signalling in this study. Ten of the 14 MAPK pathway members, *CACNA1A*, *CACNA1B*, *CACNA1C*, *CACNA1D*, *CACNA1G*, *CACNA2D1*, *CACNA2D3*, *CACNB2*, *ERBB4 and PRKCB*, are overlapped with the calcium signalling (Table 5), and 8 of them are calcium channels. Furthermore, calcium channels appear in 16 top *KEGG* pathways including glutamatergic, GABAergic, dopaminergic, cholinergic and serotonergic synapses of the ASD genes (Table 5). Our results add to the evidence that calcium and glutamatergic signalling are the significant components in ASD pathways.

Calcium signalling is a highly integral system in the human body and is increasingly shown to be implicated in ASD [104, 105]. In neurons, the arrival of the electric current induces $Ca^{2+}$ influx via voltage-gated calcium channels, and this triggers exocytosis and neurotransmitter release [106]. The voltage-gated calcium channels are tetramers containing three auxiliary subunits (β, α2δ, γ) and one pore-forming α1 subunit. Eight calcium channels (*CACNA1A*, *CACNA1B*, *CACNA1C*, *CACNA1D*, *CACNA1G*, *CACNA2D1*, *CACNA2D3*, *CACNB2*) are present in our 291 ASD geneset. Experiments using fibroblasts from monogenic [107] and non-syndromic autistic subjects [108] demonstrate aberrant calcium signalling mediated by I3PR. In

**Table 5. Converging ASD candidate genes on E/I balance and calcium signalling pathway.**

| Term ID | Term description (Background Gene Count) | ASD genes | FDR | Matching proteins in the network (IDs) |
|---|---|---|---|---|
| hsa04724 | **Glutamatergic synapse (112)** | 12 (CNS) | 2.46E-11 | *CACNA1A, CACNA1D*, DLGAP1, GRIA1, GRIK2, *GRIN1, GRIN2A*, GRIN2B, *GRM1*, HOMER1, SHANK2, SLC1A2 |
| | | 6 (CNS+PT) | 0.0179 | *CACNA1C*, DLG4, GRIK5, *ITPR1, PLCB1, PRKCB* |
| | | 19 (Com) | 1.21E-10 | *CACNA1A, CACNA1C, CACNA1D*, DLG4, DLGAP1, GRIA1, GRIK2, GRIK5, *GRIN1, GRIN2A*, GRIN2B, *GRM1*, HOMER1, SHANK3, *ITPR1, PLCB1, PRKCB*, SHANK2, SLC1A2 |
| hsa04727 | **GABAergic synapse (88)** | 11 (CNS) | 3.44E-11 | *CACNA1A, CACNA1B, CACNA1D*, GABBR2, GABRA1, GABRA3, GABRA4, GABRA5, GABRB3, GAD1, SLC12A5 |
| | | 15 (Com) | 2.48E-09 | *CACNA1A, CACNA1B, CACNA1C, CACNA1D*, GABBR2, GABRA1, GABRA3, GABRA4, GABRA5, GABRB3, GAD1, GPHN, *PRKCB*, SLC12A5, SLC6A1 |
| hsa04728 | **Dopaminergic synapse (128)** | 8 (CNS) | 2.16E-06 | *CACNA1A, CACNA1B, CACNA1D*, CAMK2A, GRIA1, *GRIN2A*, GRIN2B, SCN1A |
| | | 7 (CNS+PT) | 0.0126 | *CACNA1C*, CAMK2B, GSK3B, *ITPR1*, KIF5C, *PLCB1, PRKCB* |
| | | 15 (Com) | 7.84E-08 | *CACNA1A, CACNA1B, CACNA1C, CACNA1D, CAMK2A, CAMK2B*, GRIA1, *GRIN2A*, GRIN2B, GSK3B, *ITPR1*, KIF5C, *PLCB1, PRKCB*, SCN1A |
| hsa04725 | **Cholinergic synapse (111)** | 6 (CNS) | 0.0001 | *CACNA1A, CACNA1B, CACNA1D, CAMK2A*, KCNQ2, KCNQ3 |
| | | 11 (Com) | 2.36E-05 | *CACNA1A, CACNA1B, CACNA1C, CACNA1D, CAMK2A, CAMK2B, ITPR1*, KCNQ2, KCNQ3, *PLCB1, PRKCB* |
| hsa04726 | **Serotonergic synapse (112)** | 5 (CNS) | 0.0011 | *CACNA1A, CACNA1B, CACNA1D*, GABRB3, KCND2 |
| | | 10 (Com) | 0.00062 | *CACNA1A, CACNA1B, CACNA1C, CACNA1D*, GABRB3, *ITPR1*, KCND2, *PLCB1, PRKCB* |
| hsa04720 | **Long-term potentiation (64)** | 6 (CNS) | 6.89E-06 | *CAMK2A*, GRIA1, *GRIN1, GRIN2A*, GRIN2B, *GRM1* |
| | | 6 (CNS+PT) | 0.0025 | *CACNA1C, CAMK2B*, CREBBP, *ITPR1, PLCB1, PRKCB* |
| | | 12 (Com) | 3.14E-08 | *CACNA1C, CAMK2A, CAMK2B*, CREBBP, GRIA1, *GRIN1, GRIN2A*, GRIN2B, *GRM1, ITPR1, PLCB1, PRKCB* |
| hsa04730 | **Long-term depression (60)** | 4 (CNS) | 0.0011 | *CACNA1A*, GRIA1, *GRM1, NOS1* |
| | | 7 (Com) | 0.00051 | *CACNA1A*, GRIA1, *GRM1, ITPR1, NOS1, PLCB1, PRKCB* |
| hsa04024 | **cAMP signaling pathway (195)** | 8 (CNS) | 3.21E-05 | *ATP2B2, CACNA1D, CAMK2A*, GABBR2, GRIA1, *GRIN1, GRIN2A*, GRIN2B |
| | | 7 (CNS+PT) | 0.0382 | ATP1A3, *CACNA1C, CAMK2B*, CREBBP, PDE4A, PDE4B, RAPGEF4 |
| | | 15 (Com) | 7.17E-06 | ATP1A3, *ATP2B2, CACNA1C, CACNA1D, CAMK2A, CAMK2B*, CREBBP, GABBR2, GRIA1, *GRIN1, GRIN2A*, GRIN2B, PDE4A, PDE4B, RAPGEF4 |
| hsa04010 | **MAPK signaling pathway (293)** | 6 (CNS) | 0.0105 | *CACNA1A, CACNA1B, CACNA1D, CACNA1G, CACNA2D3, ERBB4* |
| | | 14 (Com) | 0.0013 | *CACNA1A, CACNA1B, CACNA1C, CACNA1D, CACNA1G, CACNA2D1, CACNA2D3, CACNB2, ERBB4*, KIT, MAPK8IP2, NF1, *PRKCB, TAOK2* |
| hsa04020 | **Calcium signaling pathway (179)** | 11 (CNS) | 1.80E-08 | *ATP2B2, CACNA1A, CACNA1B, CACNA1D, CACNA1G, CAMK2A, ERBB4, GRIN1, GRIN2A, GRM1, NOS1* |
| | | 16 (Com) | 7.79E-07 | *ATP2B2, CACNA1A, CACNA1B, CACNA1C, CACNA1D, CACNA1G, CAMK2A, CAMK2B, ERBB4, GRIN1, GRIN2A, GRM1, ITPR1, NOS1, PLCB1, PRKCB* |
| hsa04713 | **Circadian entrainment (93)** | 9 (CNS) | 1.74E-08 | *CACNA1D, CACNA1G, CAMK2A*, GRIA1, *GRIN1, GRIN2A*, GRIN2B, *NOS1*, NOS1AP |
| | | 5 (CNS+PT) | 0.0339 | *CACNA1C, CAMK2B, ITPR1, PLCB1, PRKCB* |
| | | 14 (Com) | 1.98E-08 | *CACNA1C, CACNA1D, CACNA1G, CAMK2A, CAMK2B*, GRIA1, *GRIN1, GRIN2A*, GRIN2B, *ITPR1, NOS1*, NOS1AP, *PLCB1, PRKCB* |

(*Continued*)

**Table 5.** (Continued)

| Term ID | Term description (Background Gene Count) | ASD genes | FDR | Matching proteins in the network (IDs) |
|---------|-------------------------------------------|-----------|-----|-----------------------------------------|
| hsa04925 | **Aldosterone synthesis and secretion (93)** | 4 (CNS) | 0.005 | ***ATP2B2, CACNA1D, CACNA1G, CAMK2A*** |
| | | 3 (CNS+PT) | 0.0475 | *ATP1A3, NR3C2,* ***PRKCB*** |
| | | 12 (Com) | 9.03E-07 | *ATP1A3,* ***ATP2B2, CACNA1C, CACNA1D, CACNA1G, CAMK2A, CAMK2B,*** *DAGLA,* ***ITPR1,*** *NR4A2,* ***PLCB1, PRKCB*** |
| hsa04723 | **Retrograde endocannabinoid signalling (148)** | 11 (CNS) | 3.77E-09 | ***CACNA1A, CACNA1B, CACNA1D,*** *GABRA1, GABRA3, GABRA4, GABRA5, GABRB3, GRIA1,* ***GRM1,*** *RIMS1* |
| | | 6 (CNS+PT) | 0.0382 | ***CACNA1C,*** *CNR1, DAGLA,* ***ITPR1, PLCB1, PRKCB*** |
| | | 17 (Com) | 1.58E-08 | ***CACNA1A, CACNA1B, CACNA1C, CACNA1D,*** *CNR1, DAGLA, GABRA1, GABRA3, GABRA4, GABRA5, GABRB3, GRIA1,* ***GRM1, ITPR1, PLCB1, PRKCB,*** *RIMS1* |
| hsa04310 | **Wnt signaling pathway (143)** | 10 (CNS+PT) | 0.00031 | *APC,* ***CAMK2B,*** *CHD8, CREBBP, CSNK1E, GSK3B,* ***PLCB1,*** *PRICKLE1, PRICKLE2,* ***PRKCB*** |
| | | 15 (Com) | 0.00017 | *ATP1A3,* ***ATP2B2, CACNA1C, CACNA1D, CAMK2A, CAMK2B,*** *CREBBP, GABBR2, GRIA1,* ***GRIN1, GRIN2A,*** *GRIN2B, PDE4A, PDE4B, RAPGEF4* |
| hsa04921 | **Oxytocin signaling pathway (149)** | 7 (CNS+PT) | 0.0179 | ***CACNA1C, CACNA2D1, CACNB2, CAMK2B, ITPR1, PLCB1, PRKCB*** |
| | | 10 (Com) | 0.00098 | ***CACNA1C, CACNA1D, CACNA2D1, CACNA2D3, CACNB2, CAMK2A, CAMK2B, ITPR1, PLCB1, PRKCB*** |
| hsa04911 | **Insulin secretion (84)** | 9 (CNS+PT) | 7.16E-05 | *TP1A3,* ***CACNA1C, CAMK2B,*** *KCNMA1,* ***PLCB1, PRKCB,*** *RAPGEF4, SNAP25, STX1A* |
| | | 11 (Com) | 2.51E-06 | *ATP1A3,* ***CACNA1C, CACNA1D, CAMK2A, CAMK2B,*** *KCNMA1,* ***PLCB1, PRKCB,*** *RAPGEF4, SNAP25, STX1A* |

addition, mutations of calcium channels have been found in ASD, for example, *CACNA1A* rs7249246/rs12609735 were associated with Chinese Han ASD [109], and *CACNA1A* mutations in Epileptic Encephalopathy [110, 111], gain of function of *CACNA1C* in Timothy syndrome with ASD [112] and recurring CNVs of *CACNA2D3* [29, 36, 113, 114]. Exome sequencing has identified various mutations of *CACNA1D* in ASD [14, 16, 29, 36, 115], epilepsy [116] developmental delay [117] endocrine issues [117, 118], *CACNA2D1* in epilepsy and intellectual disability [119] and *CACNB2* mutations in ASD [120, 121]. *CACNA1C* (rs1024582) and *CACNB2* (rs2799573) polymorphisms were suggested as the common risks across seven brain diseases [122]. Therefore, dysregulated calcium and synaptic signalling could be a commonly perturbed pathway in ASD and frequently occurring comorbidities.

Calcium channels are coupled to neuronal transmission, and E/I imbalance was proposed as a common ASD pathway previously [123]. For example, increased calcium signalling is found in $NRXN1\alpha^{+/-}$ neurons derived from ASD induced pluripotent stem cells with increased expression of voltage-gated calcium channels [124]. In the current study, KEGG pathways show that *CACNA1A, CACNA1C, CACNA1D* are involved in glutamatergic synapse; *CACNA1A, CACNA1B* and *CACNA1D* in cholinergic synapse; and *CACNA1A, CACNA1B, CACNA1C* and *CACNA1D* in GABAergic, dopaminergic, and serotonergic synapses. The glutamatergic and GABAergic transmission are the major excitatory and inhibitory pathways in the CNS, which are recurrently featured in "*Biological Processes*", "*Molecular Functions*", "*Cell Components*" and *KEGG* pathways of the 291 ASD candidate genes. Various glutamatergic receptor genes (*GRIA1, GRIK2, GRIK5, GRIN1, GRIN2A, GRIN2B, GRM1, GRM5, GRM7*), scaffolding components of synapses (*SHANK2, SHANK3, DLG4, DLGAP1*) and transport of glutamate (*SLC1A2*) were all involved in ASD and other comorbidities [110, 125–130].

Remarkably, various calcium signalling members are commonly appeared in other pathways that are perturbed in ASD (Fig 5C–5D, Table 5). For example, 11/14 molecules in Circadian entrainment, 10/14 in MAPK signalling, 7/15 in Wnt signalling, 10/10 in Oxytocin signalling, 9/12 in Aldosterone synthesis and secretion, 8/17 in Retrograde endocannabinoid signalling, 6/11 in insulin secretion, were found in calcium signalling pathway, which could be of great interest given their role in the pancreas [131]. In the top KEGG pathways identified from the CNS geneset (CNS), the CNS+PT geneset and combined (Com) 292 ASD candidates, calcium signalling members (bold) appeared in all other KEGG pathways. Therefore, calcium signalling is likely to play a major role not only in ASD but also in ASD comorbidities.

## Limitations

Like other meta-analyses, biases and limitations should be considered in pathway analyses. We first assumed that any significant ASD candidate genes shall appear in 4/5 independent datasets. However, some strong candidate genes, such as *FMR1* and *PTEN*, did not appear in the final 291 gene list. While they have much evidence to support their role in ASD, the genes did not overlap due to differences in ranking criteria of different systems. For example, PTEN was in the top ranking in SFARI, Zhang and EXAC, however it was not shortlisted from Duda's and Krishnan's score systems. Likewise, *FMR1* was ranked top in Duda's, Zhang's and SFARI, but did not pass the threshold in EXAC or Krishnan's dataset. The biological influence of *FMR1* and *PTEN* in ASD is very significant. For example, FMR1 is known to target 126/291 ASD candidates in the current study, and *PTEN* is a target of another strong ASD candidate, *CHD8*, on our list.

We also filtered out 200 genes (S26 Table), which were overlapped by four independent scoring systems but not existed in the SFARI database. 66 of them are targets of the *FMR1*, and 36 are targets of *CHD8*. Some of them like *BCL11B*, *DPF1*, *ETV1*, *NFASC*, *PAK7*, *PLXNC1*, *SCN3A*, *SLC17A7* and *SLITRK1* were also dysregulated in cerebral organoids derived from autistics with large head circumference [50], while others (*NEDD4L*, *RICTOR*, *SLC8A1*, *CELF2*, *MLLT3*, *PPFIA2*, *EPHA7*, *LRRC4C* and *RORB*) were identified from very recent single cell sequencing of autism cortical tissue [68]. This suggests they would still be modulated as downstream targets in some cases of ASD.

In the current study we hypothesized that a strong ASD candidate gene should be highly expressed in the brain and/or relative PT with strong comorbidity. However, this assumption could be potentially challenged by the following scenario. (1) If an ASD gene were development-specific but had low abundance in adult post-mortem tissues, it would be filtered out by GTEx dataset derived from donors of 20–79 age; (2) If a gene were highly expressed in a small subset of specific nuclei of the brain, which might appear not abundant in the total brain RNA; (3) If an ASD gene were modulated by ethnic genetic background, as 85.2% of GTEx donors were Caucasian European subjects, 12.7% were African-American, 1.1% were Asian and 0.3% were American-Indian. As for protein expression, the HPA is an evolving database, and genes with no protein expression data now can be updated in future releases. The same goes for cell types in tissue, which will also be incomplete, given the myriad cell types that are present in all organs. The use of single cell sequencing technology and flow cytometry will be useful in addressing the issue [132, 133].

It is interesting that many of the ASD genes are found to have sex-bias expression and/or splicing, which may be associated with sex-bias diagnosis of ASD [134–136]. We are limited in understanding its biological base by the lack of transcriptomic analysis of ASD genes. Many of the transcriptomic studies have focused on male subjects, or the ratio of female samples are too few to illustrate any sex-specific effects in ASD. It is suggested that future transcriptomic

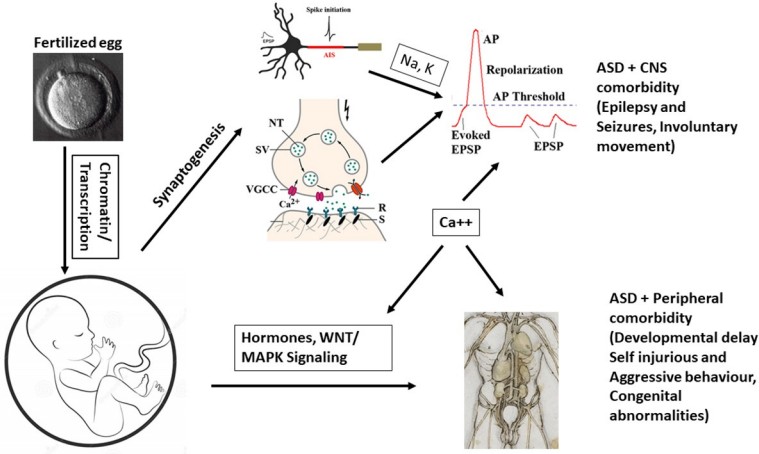

**Fig 10. Working hypothesis of ASD.** From fertilization to full development, mutations in chromatin modelling and transcription factors can contribute to altered developmental trajectory in brain formation, synaptogenesis, and organogenesis. Alterations in synaptic and ion channel genes may lead to perturbed action potentials and imbalance of E/I synapses that can contribute to the core ASD symptoms and CNS comorbidity such as epilepsy and motor functions. However, alterations in cellular, hormonal signalling and gene regulation can contribute to peripheral co-morbidities such as altered facial phenotypes and behaviour. Calcium signalling appears acting as a hub among the CNS and peripheral pathways.

studies should incorporate an even number of male and female subjects in both groups of case and control. Organoid cell lines may also be created from both sexes to make up for this short coming.

It would be desirable if gene expression datasets are available to compare from control and ASD patients at different time points not just in the brain, but also across the entire body. A recent publication [137] has proposed a paediatric cell atlas, which would collate and characterise gene expression at the single cell level across multiple tissues and time points of human development from birth to adulthood. This would be a fantastic initiative if it fully goes ahead, as such a resource would bring great insight into the co-morbidities associated with ASD. Single-cell expression data from cortical tissue of ASD subjects has become available to researchers recently [68], which could be a good starting point to analyse cell-types of interest and explore ASD heterogeneity. The availability of iPSCs from different ASD cases allows to culture and analyse different cell types in both CNS and peripheral organs, which are not easily accessed by conventional methodologies. This can be useful to explore how ASD genes may influence the biological processes during brain development, neuronal function, as well as cells of comorbidity peripheral organs.

## Conclusion

By utilizing multiple scoring systems, we have identified recurrent ASD candidate genes, with convergence on multiple pathways and processes involved in ASD (Fig 10). The use of GTEx and HPA data also gives a glimpse into their body-wide expression patterns, which has not been explored previously using ASD gene lists, which we have done so in this study. The bioinformatic analyses of CNS-specific and/or CNS+PT candidate genesets enable us to pinpoint CNS development, E/I balance and calcium signalling as important pathways involved in not just ASD but also brain comorbidity such as epilepsy. The analysis of CNS+PT geneset suggests chromosomal organisation/transcription regulation, calcium-interconnected MARK/WNT and secretion as major pathways with disruptive behaviour, developmental delay as well

as congenital abnormalities. Calcium signalling is highly interconnected amongst pathways, which could be informative in exploring complications and co-morbidities associated with ASD where calcium signalling could be involved, especially those subsets of autistic individuals who harbour mutations in these genes that can result in channelopathies.

## Supporting information

**S1 Fig. ASD genes are present in tissue-specific interaction networks.** Yellow nodes highlight ASD genes, green are tissue-specific genes, red edges are interactions between ASD genes and other partners. The graphs are in order of appearance; Adrenal Gland, Adipose subcutaneous, Artery aorta, Artery coronary, Artery tibial, Colon sigmoid, Colon transverse, Esophagus-mucosa, Esophagus-muscularis, Heart-left atrial appendage, Heart-left ventricle, Kidney, Lung, Muscle-skeletal, Pancreas, Pituitary, Small Intestine-terminal ileum, Spleen, Stomach. (TIF)

**S2 Fig. ASD genes are present in tissue-specific interaction networks.** Yellow nodes highlight ASD genes, green are tissue-specific genes, red edges are interactions between ASD genes and other partners. The graphs are in order of appearance; Adrenal Gland, Adipose subcutaneous, Artery aorta, Artery coronary, Artery tibial, Colon sigmoid, Colon transverse, Esophagus-mucosa, Esophagus-muscularis, Heart-left atrial appendage, Heart-left ventricle, Kidney, Lung, Muscle-skeletal, Pancreas, Pituitary, Small Intestine-terminal ileum, Spleen, Stomach. (TIF)

**S3 Fig. ASD genes are present in tissue-specific interaction networks.** Yellow nodes highlight ASD genes, green are tissue-specific genes, red edges are interactions between ASD genes and other partners. The graphs are in order of appearance; Adrenal Gland, Adipose subcutaneous, Artery aorta, Artery coronary, Artery tibial, Colon sigmoid, Colon transverse, Esophagus-mucosa, Esophagus-muscularis, Heart-left atrial appendage, Heart-left ventricle, Kidney, Lung, Muscle-skeletal, Pancreas, Pituitary, Small Intestine-terminal ileum, Spleen, Stomach. (PNG)

**S4 Fig. ASD genes are present in tissue-specific interaction networks.** Yellow nodes highlight ASD genes, green are tissue-specific genes, red edges are interactions between ASD genes and other partners. The graphs are in order of appearance; Adrenal Gland, Adipose subcutaneous, Artery aorta, Artery coronary, Artery tibial, Colon sigmoid, Colon transverse, Esophagus-mucosa, Esophagus-muscularis, Heart-left atrial appendage, Heart-left ventricle, Kidney, Lung, Muscle-skeletal, Pancreas, Pituitary, Small Intestine-terminal ileum, Spleen, Stomach. (TIF)

**S5 Fig. ASD genes are present in tissue-specific interaction networks.** Yellow nodes highlight ASD genes, green are tissue-specific genes, red edges are interactions between ASD genes and other partners. The graphs are in order of appearance; Adrenal Gland, Adipose subcutaneous, Artery aorta, Artery coronary, Artery tibial, Colon sigmoid, Colon transverse, Esophagus-mucosa, Esophagus-muscularis, Heart-left atrial appendage, Heart-left ventricle, Kidney, Lung, Muscle-skeletal, Pancreas, Pituitary, Small Intestine-terminal ileum, Spleen, Stomach. (TIF)

**S6 Fig. ASD genes are present in tissue-specific interaction networks.** Yellow nodes highlight ASD genes, green are tissue-specific genes, red edges are interactions between ASD genes and other partners. The graphs are in order of appearance; Adrenal Gland, Adipose subcutaneous, Artery aorta, Artery coronary, Artery tibial, Colon sigmoid, Colon transverse, Esophagus-mucosa, Esophagus-muscularis, Heart-left atrial appendage, Heart-left ventricle, Kidney,

Lung, Muscle-skeletal, Pancreas, Pituitary, Small Intestine-terminal ileum, Spleen, Stomach. (TIF)

**S7 Fig. ASD genes are present in tissue-specific interaction networks.** Yellow nodes highlight ASD genes, green are tissue-specific genes, red edges are interactions between ASD genes and other partners. The graphs are in order of appearance; Adrenal Gland, Adipose subcutaneous, Artery aorta, Artery coronary, Artery tibial, Colon sigmoid, Colon transverse, Esophagus-mucosa, Esophagus-muscularis, Heart-left atrial appendage, Heart-left ventricle, Kidney, Lung, Muscle-skeletal, Pancreas, Pituitary, Small Intestine-terminal ileum, Spleen, Stomach. (TIF)

**S8 Fig. ASD genes are present in tissue-specific interaction networks.** Yellow nodes highlight ASD genes, green are tissue-specific genes, red edges are interactions between ASD genes and other partners. The graphs are in order of appearance; Adrenal Gland, Adipose subcutaneous, Artery aorta, Artery coronary, Artery tibial, Colon sigmoid, Colon transverse, Esophagus-mucosa, Esophagus-muscularis, Heart-left atrial appendage, Heart-left ventricle, Kidney, Lung, Muscle-skeletal, Pancreas, Pituitary, Small Intestine-terminal ileum, Spleen, Stomach. (TIF)

**S9 Fig. ASD genes are present in tissue-specific interaction networks.** Yellow nodes highlight ASD genes, green are tissue-specific genes, red edges are interactions between ASD genes and other partners. The graphs are in order of appearance; Adrenal Gland, Adipose subcutaneous, Artery aorta, Artery coronary, Artery tibial, Colon sigmoid, Colon transverse, Esophagus-mucosa, Esophagus-muscularis, Heart-left atrial appendage, Heart-left ventricle, Kidney, Lung, Muscle-skeletal, Pancreas, Pituitary, Small Intestine-terminal ileum, Spleen, Stomach. (TIF)

**S10 Fig. ASD genes are present in tissue-specific interaction networks.** Yellow nodes highlight ASD genes, green are tissue-specific genes, red edges are interactions between ASD genes and other partners. The graphs are in order of appearance; Adrenal Gland, Adipose subcutaneous, Artery aorta, Artery coronary, Artery tibial, Colon sigmoid, Colon transverse, Esophagus-mucosa, Esophagus-muscularis, Heart-left atrial appendage, Heart-left ventricle, Kidney, Lung, Muscle-skeletal, Pancreas, Pituitary, Small Intestine-terminal ileum, Spleen, Stomach. (TIF)

**S11 Fig. ASD genes are present in tissue-specific interaction networks.** Yellow nodes highlight ASD genes, green are tissue-specific genes, red edges are interactions between ASD genes and other partners. The graphs are in order of appearance; Adrenal Gland, Adipose subcutaneous, Artery aorta, Artery coronary, Artery tibial, Colon sigmoid, Colon transverse, Esophagus-mucosa, Esophagus-muscularis, Heart-left atrial appendage, Heart-left ventricle, Kidney, Lung, Muscle-skeletal, Pancreas, Pituitary, Small Intestine-terminal ileum, Spleen, Stomach. (TIF)

**S12 Fig. ASD genes are present in tissue-specific interaction networks.** Yellow nodes highlight ASD genes, green are tissue-specific genes, red edges are interactions between ASD genes and other partners. The graphs are in order of appearance; Adrenal Gland, Adipose subcutaneous, Artery aorta, Artery coronary, Artery tibial, Colon sigmoid, Colon transverse, Esophagus-mucosa, Esophagus-muscularis, Heart-left atrial appendage, Heart-left ventricle, Kidney, Lung, Muscle-skeletal, Pancreas, Pituitary, Small Intestine-terminal ileum, Spleen, Stomach. (TIF)

**S13 Fig. ASD genes are present in tissue-specific interaction networks.** Yellow nodes highlight ASD genes, green are tissue-specific genes, red edges are interactions between ASD genes and other partners. The graphs are in order of appearance; Adrenal Gland, Adipose subcutaneous, Artery aorta, Artery coronary, Artery tibial, Colon sigmoid, Colon transverse, Esophagus-mucosa, Esophagus-muscularis, Heart-left atrial appendage, Heart-left ventricle, Kidney, Lung, Muscle-skeletal, Pancreas, Pituitary, Small Intestine-terminal ileum, Spleen, Stomach. (TIF)

**S14 Fig. ASD genes are present in tissue-specific interaction networks.** Yellow nodes highlight ASD genes, green are tissue-specific genes, red edges are interactions between ASD genes and other partners. The graphs are in order of appearance; Adrenal Gland, Adipose subcutaneous, Artery aorta, Artery coronary, Artery tibial, Colon sigmoid, Colon transverse, Esophagus-mucosa, Esophagus-muscularis, Heart-left atrial appendage, Heart-left ventricle, Kidney, Lung, Muscle-skeletal, Pancreas, Pituitary, Small Intestine-terminal ileum, Spleen, Stomach. (TIF)

**S15 Fig. ASD genes are present in tissue-specific interaction networks.** Yellow nodes highlight ASD genes, green are tissue-specific genes, red edges are interactions between ASD genes and other partners. The graphs are in order of appearance; Adrenal Gland, Adipose subcutaneous, Artery aorta, Artery coronary, Artery tibial, Colon sigmoid, Colon transverse, Esophagus-mucosa, Esophagus-muscularis, Heart-left atrial appendage, Heart-left ventricle, Kidney, Lung, Muscle-skeletal, Pancreas, Pituitary, Small Intestine-terminal ileum, Spleen, Stomach. (TIF)

**S16 Fig. ASD genes are present in tissue-specific interaction networks.** Yellow nodes highlight ASD genes, green are tissue-specific genes, red edges are interactions between ASD genes and other partners. The graphs are in order of appearance; Adrenal Gland, Adipose subcutaneous, Artery aorta, Artery coronary, Artery tibial, Colon sigmoid, Colon transverse, Esophagus-mucosa, Esophagus-muscularis, Heart-left atrial appendage, Heart-left ventricle, Kidney, Lung, Muscle-skeletal, Pancreas, Pituitary, Small Intestine-terminal ileum, Spleen, Stomach. (TIF)

**S17 Fig. ASD genes are present in tissue-specific interaction networks.** Yellow nodes highlight ASD genes, green are tissue-specific genes, red edges are interactions between ASD genes and other partners. The graphs are in order of appearance; Adrenal Gland, Adipose subcutaneous, Artery aorta, Artery coronary, Artery tibial, Colon sigmoid, Colon transverse, Esophagus-mucosa, Esophagus-muscularis, Heart-left atrial appendage, Heart-left ventricle, Kidney, Lung, Muscle-skeletal, Pancreas, Pituitary, Small Intestine-terminal ileum, Spleen, Stomach. (TIF)

**S18 Fig. ASD genes are present in tissue-specific interaction networks.** Yellow nodes highlight ASD genes, green are tissue-specific genes, red edges are interactions between ASD genes and other partners. The graphs are in order of appearance; Adrenal Gland, Adipose subcutaneous, Artery aorta, Artery coronary, Artery tibial, Colon sigmoid, Colon transverse, Esophagus-mucosa, Esophagus-muscularis, Heart-left atrial appendage, Heart-left ventricle, Kidney, Lung, Muscle-skeletal, Pancreas, Pituitary, Small Intestine-terminal ileum, Spleen, Stomach. (TIF)

**S19 Fig. ASD genes are present in tissue-specific interaction networks.** Yellow nodes highlight ASD genes, green are tissue-specific genes, red edges are interactions between ASD genes and other partners. The graphs are in order of appearance; Adrenal Gland, Adipose subcutaneous, Artery aorta, Artery coronary, Artery tibial, Colon sigmoid, Colon transverse, Esophagus-

mucosa, Esophagus-muscularis, Heart-left atrial appendage, Heart-left ventricle, Kidney, Lung, Muscle-skeletal, Pancreas, Pituitary, Small Intestine-terminal ileum, Spleen, Stomach. (TIF)

**S1 Table. The shortlisted genes from the scoring systems, which can be found in the supporting excel file.**
(XLSX)

**S2 Table. The shortlisted genes from the scoring systems, which can be found in the supporting excel file.**
(XLSX)

**S3 Table. The shortlisted genes from the scoring systems, which can be found in the supporting excel file.**
(XLSX)

**S4 Table. The shortlisted genes from the scoring systems, which can be found in the supporting excel file.**
(XLSX)

**S5 Table. The shortlisted genes from the scoring systems, which can be found in the supporting excel file.**
(XLSX)

**S6 Table. Jvenn overlap data.** This can be found in the supporting excel file.
(XLSX)

**S7 Table. HPA data relating to ASD genes.** This can be found in supporting excel file.
(XLSX)

**S8 Table. The 292 ASD gene list with attached scoring information: This can be found in the supporting excel file.**
(XLSX)

**S9 Table. The expression data of the 292 genes, this is found in the supporting excel file.**
(XLSX)

**S10 Table. STRING results for the CNS, CNS+PT and all 292 genes, these are found in the supporting excel files.**
(XLSX)

**S11 Table. STRING results for the CNS, CNS+PT and all 292 genes, these are found in the supporting excel files.**
(XLSX)

**S12 Table. STRING results for the CNS, CNS+PT and all 292 genes, these are found in the supporting excel files.**
(XLSX)

**S13 Table. STRING results for the CNS, CNS+PT and all 292 genes, these are found in the supporting excel files.**
(XLSX)

**S14 Table. STRING results for the CNS, CNS+PT and all 292 genes, these are found in the supporting excel files.**
(XLSX)

**S15 Table. STRING results for the CNS, CNS+PT and all 292 genes, these are found in the supporting excel files.**
(XLSX)

**S16 Table. STRING results for the CNS, CNS+PT and all 292 genes, these are found in the supporting excel files.**
(XLSX)

**S17 Table. STRING results for the CNS, CNS+PT and all 292 genes, these are found in the supporting excel files.**
(XLSX)

**S18 Table. STRING results for the CNS, CNS+PT and all 292 genes, these are found in the supporting excel files.**
(XLSX)

**S19 Table. STRING results for the CNS, CNS+PT and all 292 genes, these are found in the supporting excel files.**
(XLSX)

**S20 Table. STRING results for the CNS, CNS+PT and all 292 genes, these are found in the supporting excel files.**
(XLSX)

**S21 Table. STRING results for the CNS, CNS+PT and all 292 genes, these are found in the supporting excel files.**
(XLSX)

**S22 Table. Pathway interaction matrix, this is found in the supporting excel file.**
(XLSX)

**S23 Table. Dysregulated genes overlapped with our dataset with up/down regulation and fold changes.** This can be found in supporting excel file.
(XLSX)

**S24 Table. Enrichment Results from CSEA analysis of genes in different brain regions across multiple timepoints, this is found in the supporting excel file.**
(XLSX)

**S25 Table. Gene lists for sex-biased expression in male and female prenatal brain, sex-biased splicing, and psychiatric conditions from Psygenet, and peripheral conditions from Harmonizome database.** These are found in the supporting excel file.
(XLSX)

**S26 Table. Table of non-SFARI Overlapping genes.** This is found in the supporting excel file.
(XLSX)

**S1 File. The web links to full WebGestalt results are attached as html documents.**
(ZIP)

**S2 File. GeneOverlap code final.**
(R)

**S3 File. HPA-code final.**
(R)

## Acknowledgments

The Authors wish to thank all those who contributed to this manuscript.

## Author Contributions

**Conceptualization:** Jamie Reilly.

**Formal analysis:** Jamie Reilly.

**Supervision:** Sanbing Shen.

**Visualization:** Jamie Reilly, Sanbing Shen.

**Writing – original draft:** Jamie Reilly.

**Writing – review & editing:** Louise Gallagher, Geraldine Leader, Sanbing Shen.

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
