## [Decision Letter · Decision Letter 0]

4 Sep 2020

PONE-D-20-23352

Coupling of autism genes to tissue-wide expression and dysfunction of synapse, calcium signalling and transcriptional regulation

PLOS ONE

Dear Dr. Reilly,

Thank you for submitting your manuscript to PLOS ONE. After careful consideration, we feel that it has merit but does not fully meet PLOS ONE’s publication criteria as it currently stands. Therefore, we invite you to submit a revised version of the manuscript that addresses the points raised during the review process.

The manuscript has been assessed by two experts in the field; please find their comments appended at the end of this email. The reviewers have identified several issues, all of which needed to be addressed in a revised version of the manuscript. In particular, I would advise you to pay attention to major comments from Reviewers.

If you will need more time than this to complete your revisions, please reply to this message or contact the journal office at plosone@plos.org. Please include the following items when submitting your revised manuscript:

We look forward to receiving your revised manuscript.

Kind regards,

Nirakar Sahoo, PhD

Academic Editor

PLOS ONE

Journal Requirements:

Reviewers' comments:

Reviewer's Responses to Questions

**Comments to the Author**

1. Is the manuscript technically sound, and do the data support the conclusions?

Reviewer #1: Yes

Reviewer #2: Partly

2. Has the statistical analysis been performed appropriately and rigorously? 

Reviewer #1: Yes

Reviewer #2: Yes

3. Have the authors made all data underlying the findings in their manuscript fully available?

Reviewer #1: Yes

Reviewer #2: Yes

4. Is the manuscript presented in an intelligible fashion and written in standard English?

Reviewer #1: Yes

Reviewer #2: Yes

5. Review Comments to the Author

Reviewer #1: In this manuscript, Reilly et. al. undertake bioinformatics analysis of susceptible genes in autism spectrum disorder (ASD) and associated co-morbidities. ASD is a genetically complex neurodevelopmental syndrome, with a wide array of central nervous system (CNS) and peripheral genes being implicated. A systematic assignment of genes responsible for comorbidities is lacking. An effort toward this direction can aid in deciphering the underlying genetic determinants and mechanisms, and could be relevant clinically. To that end, the authors analyzed five publicly available ASD databases, namely, SFARI, EXAC, Krishnan, Duda and Zhang. Based on overlap of the genes in these databases and expression profile, 292 genes were finally shortlisted for further analyses. These genes could be clustered into two sets – one expressed in the CNS and the other in the CNS as well as in peripheral tissues (PT). While both genesets included genes belonging to neuronal development, ion transport and synapse, the CNS+PT set intriguingly included additional genes related to chromatin organization and gene expression.

The study represents a thorough and systematic analyses of the genes implicated in ASD using available datasets. The analysis provides a platform for further experiments looking into roles of various genes especially the ones related to chromatin organization.

A couple of issues that the authors need to address:

1) It would further strengthen the analysis if the authors provide a comparison of expression profile of the genes analyzed in normal vs ASD patients. The authors could provide the comparison for whole or a subset of genes based on availability of data. This would help in narrowing down the focus on genes whose expressions vary in diseased state for future experimental studies.

2) ASD is more prevalent in males compared to females and is also characterized by an early onset with respect to age of an individual. So, it would again strengthen the manuscript if the authors analyze expression pattern in males vs. females and whether there is any difference with age. In case there is a notable difference, then the data should be presented whereas if there is no skewed behavior, that need to be mentioned appropriately in the results.

Reviewer #2: In the present manuscript Reilly and co-authors analyze the existing gene expression data from autism spectrum disorder (ASD) samples. The authors presumed that dysregulation of a set of genes driving ASD should be visible in different scoring systems. By combining and reanalyzing the results of several scoring systems, the authors generated. list of genes that, in their opinion, reflects a hallmark set for ASD. Several findings in this set are intriguing, esp the possible involvement of extra-CNS tissues. The paper is methodologically sound and well written. However, in this Reviewer's opinion, in the present stage, the manuscript's scope is very limited, and it is likely that its impact will not be significant.

At present, there does not seem to be a lack of candidate genes for ASD. In the present form the manuscript identifies yet another candidate set, which does not illuminate what makes ASD distinct from other neurologic diseases, or whether ASD is a set of conditions. The manuscript would benefit from an informative comparison of the author's set with other diseases, identifying unique features of ASD.

As the authors indicate, it is likely that the ASD-driving dysregulation(s) occur early in development and thus, the utility of the present set is unclear.

6. PLOS authors have the option to publish the peer review history of their article (what does this mean?). If published, this will include your full peer review and any attached files.

Reviewer #1: No

Reviewer #2: No

---

## [Author Response · Author response to Decision Letter 0]

21 Oct 2020

Dear Editors, 

thank you and the reviewers for the kind remarks and constructive comments. We have addressed the points raised by the reviewers, with point-by-point response below. We have revised the manuscript with additional data which significantly strengthen the manuscript. We hope the revised manuscript can be accepted for publication on PLOS One.

Kind regards

Reviewer 1 questions

1) It would further strengthen the analysis if the authors provide a comparison of expression profile of the genes analyzed in normal vs ASD patients. The authors could provide the comparison for whole or a subset of genes based on availability of data. This would help in narrowing down the focus on genes whose expressions vary in diseased state for future experimental studies.

Response : We thank Reviewer 1 for raising this good point. We have now added a table and data on genes that were reported to be differentially expressed in various studies that overlap with our geneset. 

2) ASD is more prevalent in males compared to females and is also characterized by an early onset with respect to age of an individual. So, it would again strengthen the manuscript if the authors analyze expression pattern in males vs. females and whether there is any difference with age. In case there is a notable difference, then the data should be presented whereas if there is no skewed behavior, that need to be mentioned appropriately in the results.

Response: We have used gene lists reported by Shi et al 2016 and identified some of our ASD genes present across brain regions at prenatal stages with sex-bias expression (Fig. 8A and B). We also used CSEA webtool and found our geneset is enriched across various developmental stages in the human brain (Fig 8D). We also used GeneOveralp in Bioconductor to test significance of overlap in male biased and female biased prenatal gene sets, and identified 34 genes which are differentially spliced in male and female (Fig 8C).

Reviewer 2 questions

1) At present, there does not seem to be a lack of candidate genes for ASD. In the present form the manuscript identifies yet another candidate set, which does not illuminate what makes ASD distinct from other neurologic diseases, or whether ASD is a set of conditions. The manuscript would benefit from an informative comparison of the author's set with other diseases, identifying unique features of ASD.

Response: We have compared ASD geneset with genes from psychiatric conditions (via Psychgenet) and 4 peripheral conditions (Inflammatory bowel disease, arrythmia, Type 1 and Type 2 diabetes) from Harmonizome database (GWAS lists are from GWASdb dataset. We have identified subsets of genes overlapping with CNS snf peripheral conditions, as well as ASD unique genes (Fig. 9)

2) As the authors indicate, it is likely that the ASD-driving dysregulation(s) occur early in development and thus, the utility of the present set is unclear.

Response: We have used CSEA to test for enrichment of the ASD geneset in various brain regions across different timepoints of brain development and showed a high enrichment to specific regions, cortex in particular (Fig. 8D)

---

## [Decision Letter · Decision Letter 1]

10 Nov 2020

Coupling of autism genes to tissue-wide expression and dysfunction of synapse, calcium signalling and transcriptional regulation

PONE-D-20-23352R1

Dear Dr. Reilly,

We’re pleased to inform you that your manuscript has been judged scientifically suitable for publication and will be formally accepted for publication once it meets all outstanding technical requirements.

Kind regards,

Nirakar Sahoo, PhD

Academic Editor

PLOS ONE

Additional Editor Comments (optional):

Reviewers' comments:

Reviewer's Responses to Questions

**Comments to the Author**

1. If the authors have adequately addressed your comments raised in a previous round of review and you feel that this manuscript is now acceptable for publication, you may indicate that here to bypass the “Comments to the Author” section, enter your conflict of interest statement in the “Confidential to Editor” section, and submit your "Accept" recommendation.

Reviewer #1: All comments have been addressed

Reviewer #2: All comments have been addressed

2. Is the manuscript technically sound, and do the data support the conclusions?

Reviewer #1: Yes

Reviewer #2: Yes

3. Has the statistical analysis been performed appropriately and rigorously? 

Reviewer #1: Yes

Reviewer #2: Yes

4. Have the authors made all data underlying the findings in their manuscript fully available?

Reviewer #1: Yes

Reviewer #2: Yes

5. Is the manuscript presented in an intelligible fashion and written in standard English?

Reviewer #1: Yes

Reviewer #2: Yes

6. Review Comments to the Author

Reviewer #1: (No Response)

Reviewer #2: (No Response)

7. PLOS authors have the option to publish the peer review history of their article (what does this mean?). If published, this will include your full peer review and any attached files.

Reviewer #1: No

Reviewer #2: No

---

## [Editor Report · Acceptance letter]

1 Dec 2020

PONE-D-20-23352R1 

Coupling of autism genes to tissue-wide expression and dysfunction of synapse, calcium signalling and transcriptional regulation 

Dear Dr. Reilly:

I'm pleased to inform you that your manuscript has been deemed suitable for publication in PLOS ONE. Congratulations! Your manuscript is now with our production department. 

Kind regards, 

on behalf of

Dr. Nirakar Sahoo 

Academic Editor

PLOS ONE